# BRAINBERT: SELF-SUPERVISED REPRESENTATION LEARNING FOR INTRACRANIAL RECORDINGS

**Christopher Wang**[1,2], **Vighnesh Subramaniam**[1,2], **Adam Yaari**[1,2,3],
**Gabriel Kreiman**[2,3], **Boris Katz**[1,2], **Ignacio Cases**[1,2], **Andrei Barbu**[1,2]
[1]MIT CSAIL [2]CBMM [3]Boston Children's Hospital, Harvard Medical School
[1]{czw,vsub851,yaari,boris,cases,abarbu}@mit.edu
[2]gabriel.kreiman@tch.harvard.edu

## ABSTRACT

We create a reusable Transformer, BrainBERT, for intracranial field potential recordings bringing modern representation learning approaches to neuroscience. Much like in NLP and speech recognition, this Transformer enables classifying complex concepts, i.e., decoding neural data, with higher accuracy and with much less data by being pretrained in an unsupervised manner on a large corpus of unannotated neural recordings. Our approach generalizes to new subjects with electrodes in new positions and to unrelated tasks showing that the representations robustly disentangle the neural signal. Just like in NLP where one can study language by investigating what a language model learns, this approach enables investigating the brain by studying what a model of the brain learns. As a first step along this path, we demonstrate a new analysis of the intrinsic dimensionality of the computations in different areas of the brain. To construct BrainBERT, we combine super-resolution spectrograms of neural data with an approach designed for generating contextual representations of audio by masking. In the future, far more concepts will be decodable from neural recordings by using representation learning, potentially unlocking the brain like language models unlocked language.

## 1 INTRODUCTION

Methods that analyze neural recordings have an inherent tradeoff between power and explainability. Linear decoders, by far the most popular, provide explainability; if something is decodable, it is computed and available in that area of the brain. The decoder itself is unlikely to be performing the task we want to decode, instead relying on the brain to do so. Unfortunately, many interesting tasks and features may not be linearly decodable from the brain for many reasons including a paucity of annotated training data, noise from nearby neural processes, and the inherent spatial and temporal resolution of the instrument. More powerful methods that perform non-linear transformations have lower explainability: there is a danger that the task is not being performed by the brain, but by the decoder itself. In the limit, one could conclude that object class is computed by the retina using a CNN-based decoder but it is well established that the retina does not contain explicit information about objects. Self-supervised representation learning provides a balance between these two extremes. We learn representations that are generally useful for representing neural recordings, without any knowledge of a task being performed, and then employ a linear decoder.

The model we present here, BrainBERT [1], learns a complex non-linear transformation of neural data using a Transformer. Using BrainBERT, one can linearly decode neural recordings with much higher accuracy and with far fewer examples than from raw features. BrainBERT is pretrained once across a pool of subjects, and then provides off-the-shelf capabilities for analyzing new subjects with new electrode locations even when data is scarce. Neuroscientific experiments tend to have little data in comparison to other machine learning settings, making additional sample efficiency critical. Other applications, such as brain-computer interfaces can also benefit from shorter training regimes, as well as from BrainBERT's significant performance improvements. In addition, the embeddings of the neural data provide a new means by which to investigate the brain.

---

[1]https://github.com/czlwang/BrainBERT

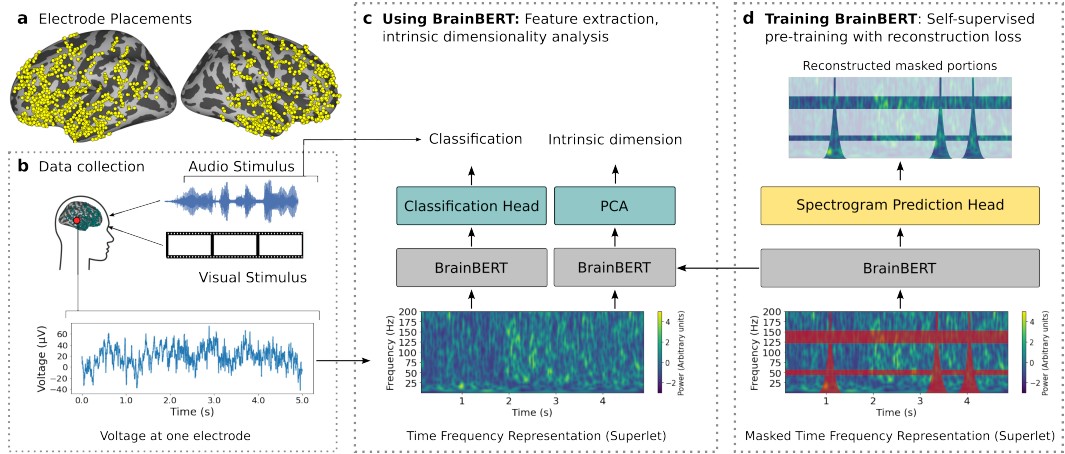

Figure 1: (a) Locations of intracranial electrodes (yellow dots) projected onto the surface of the brain across all subjects for each hemisphere. (b) Subjects watched movies while neural data was recorded (bottom, example electrode trace). (c) Neural recordings were converted to spectrograms which are embedded with BrainBERT. The resulting spectrograms are useful for many downstream tasks, like sample-efficient classification. BrainBERT can be used off-the-shelf, zero-shot, or if data is available, by fine-tuning for each subject and/or task. (d) During pretraining, BrainBERT is optimized to produce embeddings that enable reconstruction of a masked spectrogram, for which it must learn to infer the masked neural activity from the surrounding context.

BrainBERT provides contextualized neural embeddings, in the same way that masked language modeling provides contextual word embeddings. Such methods have proven themselves in areas like speech recognition where a modest amount of speech, 200 to 400 hours, leads to models from which one can linearly decode the word being spoken. We use a comparable amount of recordings, 43.7 hours across all subjects (4,551 electrode-hours), of unannotated neural recordings to build similarly reusable and robust representations.

To build contextualized embeddings, BrainBERT borrows from masked language modeling (Devlin et al., 2019) and masked audio modeling (Baevski et al., 2020; Liu et al., 2021). Given neural activity, as recorded by a stereo-electroencephalographic (SEEG) probe, we compute a spectrogram per electrode. We mask random parts of that spectrogram and train BrainBERT to produce embeddings from which the original can be reconstructed. But unlike speech audio, neural activity has fractal and scale-free characteristics (Lutzenberger et al., 1995; Freeman, 2005), meaning that similar patterns appear at different time scales and different frequencies, and identifying these patterns is often a challenge. To that end, we adapt modern neural signal processing techniques for producing super-resolution time-frequency representations of neurophysiological signals (Moca et al., 2021). Such techniques come with a variable trade-off in time-frequency resolution, which we account for in our adaptive masking strategy. Finally, the activity captured by intracranial electrodes is often sparse. To incentivize the model to better represent short-bursts of packet activity, we use a *content-aware* loss that places more weight on non-zero spectrogram elements.

Our contributions are:

1. the BrainBERT model — a reusable, off-the-shelf, subject-agnostic, and electrode-agnostic model that provides embeddings for intracranial recordings,
2. a demonstration that BrainBERT systematically improves the performance of linear decoders,
3. a demonstration that BrainBERT generalizes to previously unseen subjects with new electrode locations, and
4. a novel analysis of the intrinsic dimensionality of the computations performed by different parts of the brain made possible by BrainBERT embeddings.

## 2 METHOD

The core of BrainBERT is a stack of Transformer encoder layers (Vaswani et al., 2017). In pretraining, BrainBERT receives an unannotated time-frequency representation of the neural signal as input. This input is randomly masked, and the model learns to reconstruct the missing portions.

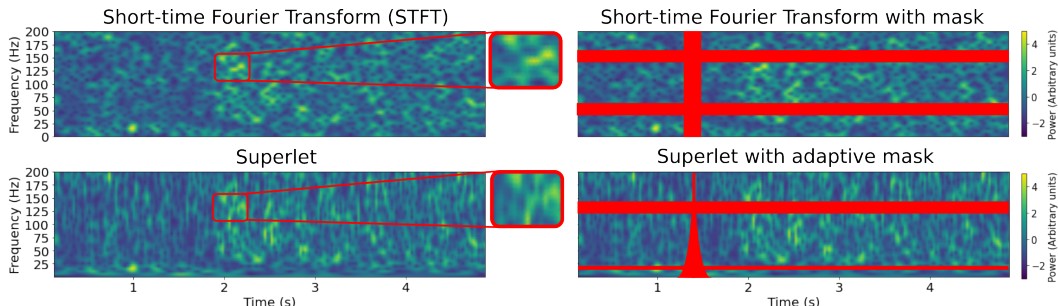

Figure 2: BrainBERT can be trained to either use spectrograms computed by a traditional method, such as the short-time Fourier Transform (top left), or modern methods designed for neural data, such as the superlet transform (bottom left). Shown above are spectrograms from a single electrode over a 5s interval. Superlets provide superresolution by compositing together Morlet wavelet transforms across a range of orders. As in Liu et al. (2021), we mask multiple continuous bands of random frequencies and time intervals (top right, red horizontal and vertical rectangles). Since the temporal resolution of superlets falls off as the inverse function of frequency (bottom right), we adopt a masking strategy that reflects this.

The pretrained BrainBERT weights can then be combined with a classification head and trained on decoding tasks using supervised data.

**Architecture**     Given the voltage measurements $x \in \mathbb{R}^{r \cdot t}$ of a single electrode sampled at rate $r$ for $t$ seconds, we first find the time-frequency representation $\Phi(x) = \mathbf{Y} \in \mathbb{R}^{n \times m}$, which has $n$ frequency channels and $m$ time frames. BrainBERT is built around a Transformer encoder stack core (Vaswani et al., 2017) with $N$ layers, each with $H$ attention heads and intermediate hidden dimension $d_h$. The inputs to the first layer are non-contextual embeddings for each time frame, $E_{\text{in}}^0 = (W_{\text{in}} Y + P)$, which are produced using a weight matrix $W_{\text{in}} \in \mathbb{R}^{d_h \times n}$ and combined with a static positional embedding $P$ (Devlin et al., 2019). Each layer applies self-attention and a feed forward layer to the input, with layer normalization (Ba et al., 2016) and dropout (Srivastava et al., 2014) being applied after each. The outputs $E_{\text{out}}^j$ of the $j$-th layer, become the inputs to the $(j+1)$-th layer. The outputs of BrainBERT are $E_{\text{out}}^N$, the outputs at the $N$-th layer.

During pretraining, the hidden-layer outputs from the top of the stack are passed as input to a spectrogram prediction head, which is a stacked linear network with a single hidden layer, GeLU activation (Hendrycks & Gimpel, 2016), and layer normalization.

**Time-frequency representations**     BrainBERT can take two different types of time-frequency representations as input: the Short-Time Fourier Transform (STFT) and the superlet transform (Moca et al., 2021), which is a composite of Morlet wavelet transforms; see appendix A.

Due to the Heisenberg-Gabor uncertainty principle, there exists an inherent trade-off between the time and frequency resolution that any representation can provide. The most salient difference between the STFT and superlet transform is the way they handle this trade-off. For the STFT, resolution is fixed for all frequencies. For the superlet transform, temporal resolution increases with frequency. This is a well-motivated choice for neural signal, where high frequency oscillations are tightly localized in time.

For both types of representations, the spectrograms are z-scored per frequency bin. This is done in order to better reveal oscillations at higher frequencies, which are usually hidden by the lower frequencies that typically dominate in power. Additionally, the z-score normalization makes Brain-BERT agnostic to the role that each frequency band might play for different tasks. Specific frequency bands have previously been implicated in different cognitive processes such as language (Babajani-Feremi et al., 2016), emotion (Drane et al., 2021), and vision (Jia et al., 2013). By intentionally putting all frequency bands on equal footing, we ensure that BrainBERT embeddings will be generic and useful for a wide variety of tasks.

**Pretraining**     During pretraining, a masking strategy is applied to the time-frequency representation $\mathbf{Y} \in \mathbb{R}^{n \times m}$, and an augmented view of the spectrogram, $\tilde{\mathbf{Y}}$, is produced. Given $\tilde{\mathbf{Y}}$, BrainBERT creates representations for a spectrogram prediction network, which produces a reconstruction $\hat{\mathbf{Y}}$ of the original signal; see fig. 1.d.

For the STFT, we adapt the masking strategy of Liu et al. (2021), in which the spectrogram is corrupted at randomly chosen time and frequency intervals. The width of each time-mask is a randomly chosen integer from the range $[\text{step}_{\min}^{\text{time}}, \text{step}_{\max}^{\text{time}}]$. Following Devlin et al. (2019), intervals selected for masking are probabilistically either left untouched (probability $p_{\text{ID}}$), replaced with a random slice of the same spectrogram (probability $p_{\text{replace}}$), or filled in with zeros otherwise; see appendix D for the pseudocode summary and parameter settings. The procedure for masking frequency intervals is similar, but the width is chosen from the range $[\text{step}_{\min}^{\text{freq}}, \text{step}_{\max}^{\text{freq}}]$.

For the superlet model, we use an adaptive masking scheme that reflects the variable trade-off in time-frequency resolution made by the continuous wavelet transform. The temporal width of the time mask increases with the inverse of frequency. Similarly, when masking frequencies, more channels are masked at higher frequencies; see appendix C for parameters.

The model is optimized according to a novel *content aware* loss. For speech audio modeling, it is typical to use an L1 reconstruction loss (Liu et al., 2020; 2021):

$$\mathcal{L}_L = \frac{1}{|M|} \sum_{(i,j) \in M} \left| \mathbf{Y}_{i,j} - \hat{\mathbf{Y}}_{i,j} \right| \tag{1}$$

where $M$ is the set of masked spectrogram positions.

But intracranial neural signal is characterized by spiking activity and short oscillation bursts. Furthermore, since the spectrogram is z-scored along the time-axis, approximately 68% of the z-scored spectrogram is 0 or $< 1$. So, with just a bare reconstruction loss, the model tends to predict 0 for most of the masked portions, especially in the early stages of pretraining. To discourage this and to speed convergence, we add a term to the loss which incentivizes the faithful reconstruction of the spectrogram elements that are $\gamma$ far away from 0, for some threshold $\gamma$.

$$\mathcal{L}_C = \frac{1}{|\{(i,j) \mid \mathbf{Y}_{i,j} > \gamma\}|} \sum_{(i,j) \mid (i,j) \in M, \mathbf{Y}_{i,j} > \gamma} \left| \mathbf{Y}_{i,j} - \hat{\mathbf{Y}}_{i,j} \right| \tag{2}$$

This incentivizes the model to faithfully represent those portions of the signal where neural processes are most likely occurring.

Then, our loss function is:

$$\mathcal{L} = \mathcal{L}_L + \alpha \mathcal{L}_C \tag{3}$$

**Fine-tuning** After pretraining, BrainBERT can be used as a feature extractor for a linear classifier. Then, given an input spectrogram $\mathbf{Y} \in \mathbb{R}^{n \times 2l}$, the features are $\mathbf{E} = \text{BrainBERT}(\mathbf{Y})$. For a window size $k$, the center $2k$ features are $\mathbf{W} = \mathbf{E}_{:, l-k:l+k}$, and the input to the classification network is the vector resulting from taking the mean of $\mathbf{W}$ along the time (first) axis. We use $k = 5$, which corresponds with a time duration of $\approx 244$ms. During training, BrainBERT's weights can either be frozen (no fine-tuning) or they can be updated along with the classification head (fine-tuning). Fine-tuning uses more compute resources, but often results in better performance. We explore both use cases in this work.

**Data** Invasive intracranial field potential recordings were collected during 26 sessions from 10 subjects (5 male, 5 female; aged 4-19, $\mu$ 11.9, $\sigma$ 4.6) with pharmacologically intractable epilepsy. Approximately, 4.37 hours of data were collected from each subject; see appendix J. During each session, subjects watched a feature length movie in a quiet room while their neural data was recorded at a rate of 2kHz. Brain activity was measured from electrodes on stereo-electroencephalographic (SEEG) probes, following the methods of Liu et al. (2009). Across all subjects, data was recorded from a total of 1,688 electrodes, with a mean of 167 electrodes per subject. Line noise was removed and the signal was Laplacian re-referenced (Li et al., 2018). During pretraining, data from all subjects and electrodes is segmented into 5s intervals, and all segments are combined into a single training pool. The complete preprocessing pipeline is described in appendix F.

## 3 EXPERIMENTS

For pretraining purposes, neural recordings from 19 of the sessions ($\mu_{\text{duration}} = 2.3$hrs) was selected, and the remaining 7 sessions ($\mu_{\text{duration}} = 2.55$hrs) were held out to evaluate performance on decoding tasks; see section 3 Tasks. All ten subjects are represented in the pretraining data. In total,

| | Sentence onset | Speech/Non-speech | Pitch | Volume | Task Avg. |
|---|---|---|---|---|---|
| Linear (.25s, time domain) | $.54 \pm .04$ | $.52 \pm .03$ | $.48 \pm .09$ | $.54 \pm .09$ | $.52 \pm .07$ |
| Linear (5s, time domain) | $.63 \pm .04$ | $.58 \pm .06$ | $.58 \pm .07$ | $.56 \pm .19$ | $.59 \pm .11$ |
| Linear (.25s, STFT) | $.60 \pm .04$ | $.53 \pm .04$ | $.51 \pm .06$ | $.52 \pm .06$ | $.54 \pm .06$ |
| Linear (.25s, superlet) | $.59 \pm .03$ | $.53 \pm .03$ | $.52 \pm .06$ | $.53 \pm .08$ | $.54 \pm .06$ |
| Deep NN (5s, 5 FF layers) | $.72 \pm .10$ | $.67 \pm .08$ | $.57 \pm .06$ | $.54 \pm .11$ | $.63 \pm .12$ |
| BrainBERT (STFT) | $\mathbf{.82 \pm .07}$ | $\mathbf{.93 \pm .03}$ | $\mathbf{.75 \pm .03}$ | $.83 \pm .09$ | $\mathbf{.83 \pm .09}$ |
|    random initialization | $.68 \pm .10$ | $.59 \pm .11$ | $.50 \pm .05$ | $.61 \pm .11$ | $.60 \pm .12$ |
|    without content aware loss | $.81 \pm .07$ | $.90 \pm .12$ | $.68 \pm .06$ | $\mathbf{.84 \pm .04}$ | $.81 \pm .11$ |
| BrainBERT (superlet) | $.78 \pm .08$ | $.86 \pm .06$ | $.62 \pm .05$ | $.70 \pm .10$ | $.74 \pm .12$ |
|    random initialization | $.66 \pm .09$ | $.54 \pm .04$ | $.52 \pm .07$ | $.60 \pm .05$ | $.58 \pm .09$ |
|    without content aware loss | $.74 \pm .12$ | $.79 \pm .14$ | $.59 \pm .05$ | $.70 \pm .13$ | $.71 \pm .14$ |
|    without adaptive mask | $.78 \pm .08$ | $.86 \pm .05$ | $.70 \pm .04$ | $.76 \pm .06$ | $.77 \pm .08$ |

Table 1: BrainBERT improves the performance of linear decoders across a wide range of tasks. The tasks vary from low-level perceptual tasks like determining the volume of the audio that the subject heard, mid-level tasks like determining if the subject is listening to speech, and high-level tasks like determining if the subject heard the start of a new sentence as opposed to some other audio of speech. Five baselines (top): four linear decoders with varying receptive fields (taking as input either 0.25s and 5s of the neural recordings) and varying input modalities (the time-domain signal or spectorgrams computed by STFT or Superlets). BrainBERT with either STFT or Superlets (bottom). A linear decoder is trained on top of the embeddings from BrainBERT, jointly tuning both the parameters of BrainBERT and the linear decoder. See Table 2 for results where BrainBERT is held fixed. Results are reported on the 10 best-performing electrodes (as measured by AUC) selected with the linear model (5s, time domain) model. This ensures that results are biased away from BrainBERT and toward the linear decoders. Despite this, BrainBERT significantly outperforms any other decoders, often by very large margins. Five different ablations are described in the text.

there are 21 unique movies represented across all sessions — the set of movies used in pretraining does not overlap with the held-out sessions. From this held-out set, for each task, session, and electrode, we extract annotated examples. Each example corresponds with a time-segment of the session recordings with time stamp $t$. Depending on the classifier architecture, the input for each example is either 0.25s or 5s of surrounding context, centered on $t$. The time-segments in each split are consistent across electrodes.

We run two sets of experiments, pretraining BrainBERT separately on both STFT and superlet representations. Pretraining examples are obtained by segmenting the neural recordings into 5s intervals.

**Tasks** To demonstrate BrainBERT's abilities to assist with a wide range of classification tasks, we demonstrate it on four such tasks. Tasks range from low-level perception such as determining the volume of the audio the subject is listening to, to mid-level tasks such as determining if the subject is hearing speech or non-speech, as well as the pitch of the overheard words, and higher-level tasks such as determining if the subject just heard the onset of a sentence as opposed to non-speech sounds. BrainBERT is pretrained without any knowledge of these tasks.

Volume of the audio was computed automatically from the audio track of the movie. Segments of neural recordings that corresponded to where audio track that was one standard deviation above or below the mean in terms of volume were selected. Speech vs. non-speech examples were manually annotated. An automatic speech recognizer first produced a rough transcript which was then corrected by annotators in the lab. Pitch was automatically computed for each word with Librosa (McFee et al., 2015) and the same one standard deviation cutoff was used to select the data. Sentence onsets were automatically derived from the transcript. In the case of the sentence onset task and the speech vs. non-speech task, the two classes were explicitly balanced. See appendix J for the exact number of examples used per class. All results are reported in terms of the ROC (receiver operating characteristic) AUC (area under the curve), where chance performance is 0.50 regardless of whether the dataset were balanced or not. The data was randomly split into a 80/10/10 training/validation set/test set.

**Baselines** To establish baselines on supervised performance, we train decoding networks both on the raw neural signal $x$ and on the time-frequency representations, $\Phi(x)$. We train two linear classifiers, at two different time scales, which take the raw neural signal as input. One network receives 250ms of input, which is approximately the same size (244ms) as the window of BrainBERT

| | Sentence onset | Speech/Non-speech | Pitch | Volume | Task Avg. |
|---|---|---|---|---|---|
| BrainBERT (STFT) | $.66 \pm .03$ | $.63 \pm .05$ | $.51 \pm .07$ | $.60 \pm .05$ | $.60 \pm .08$ |
|    random initialization | $.62 \pm .04$ | $.57 \pm .04$ | $.52 \pm .06$ | $.59 \pm .07$ | $.57 \pm .06$ |
|    without content aware loss | $.65 \pm .04$ | $.64 \pm .04$ | $.51 \pm .07$ | $.60 \pm .05$ | $.60 \pm .08$ |
| BrainBERT (superlet) | $\mathbf{.71 \pm .06}$ | $\mathbf{.69 \pm .06}$ | $.53 \pm .07$ | $.60 \pm .08$ | $\mathbf{.63 \pm .10}$ |
|    random initialization | $.62 \pm .03$ | $.56 \pm .05$ | $.52 \pm .06$ | $.59 \pm .08$ | $.57 \pm .07$ |
|    without content aware loss | $.68 \pm .06$ | $.67 \pm .07$ | $.53 \pm .07$ | $.60 \pm .07$ | $.62 \pm .09$ |
|    without adaptive mask | $.67 \pm .06$ | $.66 \pm .06$ | $\mathbf{.54 \pm .06}$ | $.60 \pm .07$ | $.62 \pm .08$ |

Table 2: BrainBERT can be used with fine-tuning or without by freezing its weights. In Table 1, we showed results that include fine-tuning, jointly tuning the linear decoder and BrainBERT. Here, we show AUC with frozen weights. BrainBERT with frozen parameters and just a linear decoder has the same performance as the deep NN shown in Table 1. We get the best of both worlds: performance of deep networks and explainability of linear decoders. Note that superlet-based BrainBERT generalizes better without fine-tuning than the STFT-based BrainBERT.

embeddings received by the classifier network. The other linear network receives 5s of raw signal, which is the size of the receptive field for the BrainBERT embeddings. These networks provide baselines on the amount of linearly decodable information available in the raw signal. To get a sense of what is possible with more expressive power, we also train a deep network, which has 5 stacked feed forward layers.

Because BrainBERT receives a time-frequency representation, $\Phi(x)$, as input, we also train two linear networks that take time-frequency representations (either STFT or superlet) as input. Complete descriptions of each network architecture is given in appendix E.

Evaluation is performed per task and per electrode. In order to keep computational costs reasonable, we only evaluate on a subset of electrodes. Our comparison to the baselines thus proceeds in two stages. First, the Linear (5s time domain input) network is trained once per electrode, for all data recorded in the held-out sessions; see section 3. Then, we select, per task, the top 10 electrodes, $S_{\text{task}}$, for which the linear decoder achieves the best ROC-AUC. For each task, this set $S_{\text{task}}$ is held fixed for all baselines, ablations, and comparisons to our model.

## 4 RESULTS

BrainBERT is pretrained as described above in an unsupervised manner without any manual annotations. Our goal is for BrainBERT to be usable off-the-shelf for new experiments with new data. To that end, we demonstrate that it improves decoding performance on held-out data and held-out electrodes. Next, we show that its performance is conserved for never-before-seen subjects with novel electrode locations. After that, we demonstrate that a linear decoder using BrainBERT embeddings outperforms the alternatives with 1/5th as much data. Finally, we show that not only are BrainBERT embeddings extremely useful practically, they open the doors to new kinds of analyses like investigating properties of the computations being carried out by different brain regions.

**Improving decoding accuracy** We compared a pretrained BrainBERT against the baseline models described above (variants of linear decoders and a 5 feed-forward layer network); see table 1. Using the linear decoder (5s time domain input) on the original data, without BrainBERT, we chose the 10 best electrodes across all of the subjects. This puts BrainBERT at a disadvantage since its highest performing electrodes might not be included in the top 10 from the linear decoder. Then we investigated if BrainBERT could offer a performance improvement when performing the four tasks with data from these 10 electrodes. Decoding BrainBERT embeddings resulted in far higher performance on every task. On average, a linear decoder using BrainBERT had an AUC of 0.83. The baseline deep network had an AUC of 0.63. A linear decoder on the unprocessed data had an AUC of 0.59. This difference is very meaningful: it takes a result which is marginal with a linear decoder and makes it an unqualified success using BrainBERT.

**Decoding accuracy without fine-tuning** In the previous experiment, we fine-tuned BrainBERT in conjunction with the linear decoder. In table 2, we report results without fine-tuning, only updating the parameters of the linear decoder. BrainBERT without fine-tuning has an average AUC of 0.63, on par with the baseline deep network. This is what we mean by BrainBERT being the best of both worlds: it provides the performance of a complex network with many non-linear transformations

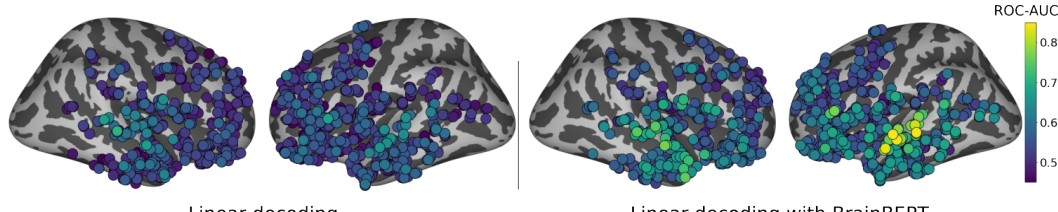

Linear decoding                           Linear decoding with BrainBERT

Figure 3: Using a linear decoder for classifying sentence onsets either (left) directly with the neural recordings or (right) with BrainBERT (superlet input) embeddings. Each circle denotes a different electrode. The color shows the classification performance (see color map on right). Electrodes are shown on the left or right hemispheres. Chance has AUC of 0.5. Only the 947 held-out electrodes are shown. Using BrainBERT highlights far more relevant electrodes, provides much better decoding accuracy, and more convincingly identifies language-related regions in the superior temporal and frontal regions.

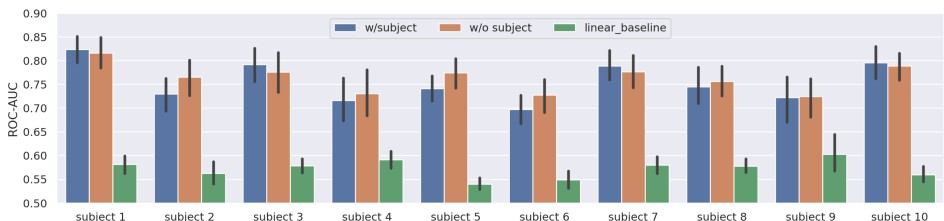

Figure 4: BrainBERT can be used off-the-shelf for new experiments with new subjects that have new electrode locations. The performance of BrainBERT does not depend on the subject data being seen during pretraining. We show AUC averaged across the four decoding tasks (table 1), in each case finetuning BrainBERT's weights and training a linear decoder. Ten held-out electrodes were chosen from the held-out subject's data. As before, these electrodes have the highest linear decoding accuracy on the original data without BrainBERT. The first two columns in each group show Brain-BERT decoding results when a given subject is included in the pretraining set (blue), and when that subject is held out (orange). The performance difference between the two is negligible, and both significantly outperform the linear decoding baseline (green), showing that BrainBERT is robust and can be used off the shelf. Error bars show a 95% confidence interval over the ten electrodes.

while only tuning a task-specific linear decoder. Note that STFT provides the best performance when fine-tuning BrainBERT while superlets appear to be more robust when not fine-tuning.

**Ablations** BrainBERT might work only because of its structure; even a random network can disentangle part of its input. To verify that the pretraining is useful, in both table 1 and table 2 we show that BrainBERT with randomly initialized weights is considerably worse at increasing decoding accuracy than the pretrained BrainBERT. For example, BrainBERT with STFT and fine-tuning gains on average 0.23 AUC over its randomized ablation on average (table 1). In BrainBERT, the weights matter, not just the structural prior.

BrainBERT has two other novelties in it: the content aware loss and the superlet adaptive mask. We evaluate the impact on both in table 1 and table 2. The content aware loss has a mild impact on performance, e.g., BrainBERT with superlets and fine-tuning gains 0.01 AUC, increasing performance in 3 out of 4 tasks by about 5%. The adaptive mask has a similar impact, e.g., on the same model it also gains 0.01 AUC increasing performance on 2 out of 4 tasks by about 5%. Cumulatively these changes have a significant impact on performance, although a second order one compared to BrainBERT itself.

**Generalizing to new subjects** Since BrainBERT is meant to be used off-the-shelf to provide a performance boost for neuroscientific experiments, it must generalize to never-before-seen subjects with novel electrode locations. To test this, we perform a hold-one-out analysis. For a given subject, we train two variants of BrainBERT with superlet input, one that includes all subjects and one where the subject is held out. We evaluate the fine-tuning performance on all of the four tasks described above with both variants. In this analysis, we include only the held-out subject's best 10 electrodes (while previously we had included the best 10 electrodes across all subjects). We find that the performance difference while generalizing to a new subject with new electrode locations is negligible; see fig. 4. For all cases, BrainBERT still gains a large boost over a linear decoder on the original data. BrainBERT generalizes extremely well to new subjects.

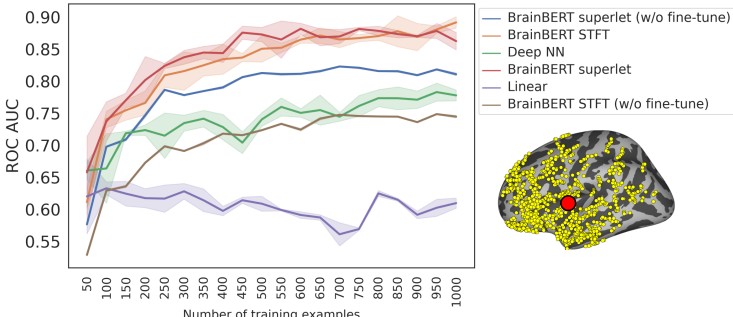

Figure 5: BrainBERT not only improves decoding accuracy, but it does so with far less data than other approaches. Performance on sentence onset classification is shown for an electrode in the superior temporal gyrus (red dot in brain inset). Error bars show standard deviation over 3 random seeds. Linear decoders (blue) saturate quickly; deep neural networks (green, 5 FF layers, details in text) perform much better but they lose explainability. BrainBERT without fine tuning matches the performance of deep networks, without needing to learn new non-linearities. With fine-tuning, BrainBERT significant outperforms, and it does so with 1/5th as many examples (the deep NN peak at 1,000 examples is exceeded with only 150 examples). This is a critical enabling step for other analyses where subjects may participate in only a few dozen trials as well as for BCI.

**Improved data efficiency**    Not only does BrainBERT increase performance, it does so with far fewer training examples than a linear decoder or the baseline deep network require. In fig. 5, we compare the decoding AUC of baseline models and BrainBERT variants as a function of training examples. All models are trained on the data of a single electrode in the left superior temporal gyrus, which was selected from a full-brain analysis (see fig. 3) to find the electrode with the best combined ranking of linear decoding (5s time duration) and BrainBERT performance. BrainBERT has a much steeper learning curve: it achieves the performance of the best baseline model with 1,000 examples with only 150 examples. In addition, if more examples are available, BrainBERT is able to use them and further increase its performance.

**Intrinsic dimension**    BrainBERT's embeddings provide value beyond just their immediate uses in increasing the performance of decoders. We demonstrate a first use of these embeddings to investigate a property of the computations in different areas of the brain without the restrictive lens of a task. We ask: what is the representation dimensionality (ID) of the neural activity? In other words, what is the minimal representation of that activity? Work with artificial networks has shown that this quantity can separate input and output regions from regions in a network that perform computations, such that high ID regions reflect increased intermediate preprocessing of the data (Ansuini et al., 2019).

Given a pretrained BrainBERT model, we estimate the intrinsic dimension at each electrode by way of principal component analysis (PCA); see appendix I. For each electrode, we find the number of PCs required to account for 95% of the variance in the signal (see fig. 6). We find that results are locally consistent, and that intrinsic dimension seems to vary fairly smoothly. To determine where high ID processing is located, we take the electrodes which fall in the top 10-th percentile. These electrodes mainly fall in the frontal and temporal lobes. Among these electrodes, the regions with the highest mean ID are the supramarginal gyrus, which is involved with phonological processing (Deschamps et al., 2014), the lateral obitofrontal cortex, involved with sensory integration (Rolls, 2004), and the amygdala, involved with emotion (Gallagher & Chiba, 1996). The intrinsic dimension is computed without respect to any decoding task, e.g. speech vs. non-speech or volume classification, and provides a novel view of functional regions in a task agnostic way.

Investigations into the embedding space of models of neural data like BrainBERT are in their infancy. They could in the future lead to an understanding of the relationship between brain regions, the general flow of computation in the brain, and brain state changes over time such as during sleep.

# 5    RELATED WORK

**Self-supervised representations**    Our work builds on masked-spectrogram modeling approaches from the field of speech processing (Liu et al., 2021; 2020; Ling & Liu, 2020), which use a reconstructive loss to learn representations of Mel-scale audio spectrograms. In other fields, self-supervised models have been used to produce representations for a variety of downstream applica-

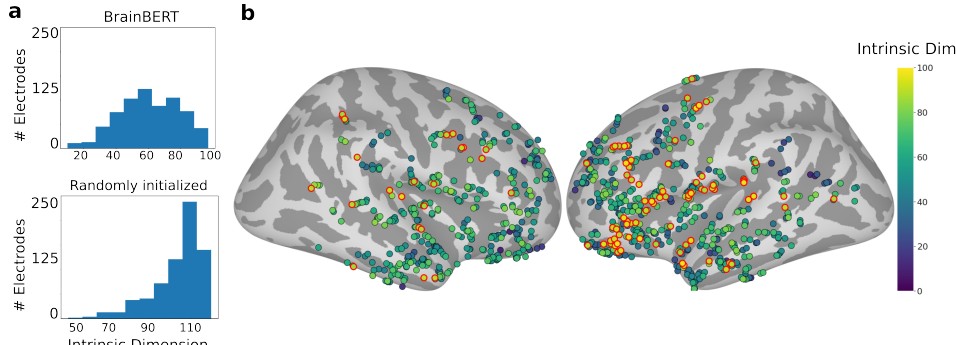

Figure 6: Given neural recordings without any annotations, we compute the intrinsic dimensonality (ID) of the BrainBERT embeddings at each electrode. (a) These embeddings lie in a lower dimensional space (top) than those produced by a randomly initialized model (bottom). (b) The electrodes with the highest ID (top 10-th percentile; circled in red) can be found mainly in the frontal and temporal lobes, and demonstrate that electrodes that participate in similar computations on similar data will have similar ID, providing a new data-driven metric by which to identify functional regions and the relationship between them.

tions (Oord et al., 2018), including language (Devlin et al., 2019), audio (Baevski et al., 2020; Hsu et al., 2021), and images (Chen et al., 2020). The representations learned by these models can in turn be used to investigate their input domains. Language models have been used to study language processing by examining the semantic and syntactic content of the embeddings (Tenney et al., 2019; Roger et al., 2022) and the dimensionality of the input space (Hernandez & Andreas, 2021). Our work opens the doors for similar studies of neural processing. Here we investigate the intrinsic dimensionality of neural computation.

**Neural decoding**    Intracranial probes, such as SEEG, have been proposed as key components in the next-generation of brain machine interfaces (Jackson & Hall, 2016; Herff et al., 2020). Our work is the first to provide a modern representation learning approach to feature extraction for intracranial signals. Previous work gives approaches for other modalities, including fMRI data (Malkiel et al., 2021), and EEG (Kostas et al., 2021; Banville et al., 2021). But the signal obtained for these modalities have fundamentally different characteristics. Importantly, they lack the temporal resolution and spatial precision of intracranial electrodes, which directly measure electrophysiological brain activity from a small population of neurons.

## 6    CONCLUSION

We present a self-supervised approach for learning representations of neurophysiological data. The resulting representations have three chief benefits. First, they improve the accuracy and data-efficiency of neural decoding. This is no small matter: many core results in neuroscience hinge on whether a linear decoder can perform a certain task. Moreover, neuroscience tends to be in the small data regime where gathering data for even small experiments is a significant investment; here, a large reduction in the amount of data required to decode something can make a large difference. Such decoding is also critical to building the next generation of brain machine interfaces. Second, they provide this performance improvement while maintaining the benefits of linear decoders, i.e., explainability, with the performance of complex decoders. This is a direct upgrade of one of the most popular techniques in neuroscience. Finally, by providing task-agnostic embeddings that can then be analyzed, they open the door for new investigations of cognitive processes. The changes in representations across time may reveal mechanisms and dynamics underlying sleep and other brain state changes in a completely data-driven manner. To further refine the representations that Brain-BERT builds, in the future, we intend to train much larger variants on continuous 24/7 recordings from numerous subjects. Fields like NLP have been revolutionized by such models; we hope that neuroscience will see similar benefits.

BrainBERT is available as a resource to the community along with the data and scripts to reproduce the results presented here. It should be noted that the pretrained weights that we release have been trained on activity induced by a particular stimulus, i.e., passive movie viewing. If the downstream use case departs significantly from this setting, then we recommend pretraining BrainBERT from scratch to improve performance. Otherwise, most  groups investigating intracranial recordings should be able to use BrainBERT as a drop-in addition to their existing pipelines.

## ACKNOWLEDGEMENTS

This work was supported by the Center for Brains, Minds, and Machines, NSF STC award CCF-1231216, the NSF award 2124052, the MIT CSAIL Systems that Learn Initiative, the MIT CSAIL Machine Learning Applications Initiative, the MIT-IBM Watson AI Lab, the CBMM-Siemens Graduate Fellowship, the DARPA Artificial Social Intelligence for Successful Teams (ASIST) program, the DARPA Knowledge Management at Scale and Speed (KMASS) program, the United States Air Force Research Laboratory and United States Air Force Artificial Intelligence Accelerator under Cooperative Agreement Number FA8750-19-2-1000, the Air Force Office of Scientific Research (AFOSR) under award number FA9550-21-1-0014, and the Office of Naval Research under award number N00014-20-1-2589 and award number N00014-20-1-2643. The views and conclusions contained in this document are those of the authors and should not be interpreted as representing the official policies, either expressed or implied, of the U.S. Government. The U.S. Government is authorized to reproduce and distribute reprints for Government purposes notwithstanding any copyright notation herein We would also like to thank Colin Conwell and Hector Luis Penagos-Vargas for helpful comments and discussion.

## ETHICS STATEMENT

Experiments that contributed to this work were approved by IRB. All subjects consented to participate. All electrode locations were exclusively dictated by clinical considerations.

## REPRODUCIBILITY STATEMENT

Code to train models and reproduce the results was submitted as part of the supplementary materials and can be accessed here: https://github.com/czlwang/BrainBERT.

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

## A  SUPERLETS

Superlet transforms are a technique for time-frequency analysis introduced by Moca et al. (2021). The superlet transform is formed from a composite Morlet wavelet transforms. In contrast to traditional time-frequency techniques, such as the Short-time Fourier Transform (STFT) which has a fixed time-frequency resolution trade-off for all frequencies, the Morlet wavelet has lower temporal resolution at lower frequencies. This variable trade-off allows the resulting representation to better capture the dynamics of neural signal, for which high frequency oscillations often occur in short bursts and low frequency oscillations persist for longer durations.

However, the Morlet transform is not Pareto optimal, and the superlet transform makes improvements to both time and frequency resolution simultaneously in order to better capture the self-similar oscillations across different frequencies which are often present in neural signal. In this section, we briefly summarize the content of Moca et al. (2021) relevant to our work.

Consider the following formulation of the Morlet wavelet for a given frequency of interest $f$:

$$\psi_{c,f}(t) = \frac{1}{B_c\sqrt{2\pi}} \exp\left(-\frac{t^2}{2B_c^2}\right) \exp\left(j2\pi ft\right) \tag{4}$$

$$B_c = \frac{c}{f} \tag{5}$$

Here, $c$ is the number of cycles, which is a parameter that controls the time-frequency resolution trade-off. The expression for the wavelet on the right side of the equation eq. (4) can be understood as being composed of two terms: the first term can be thought of as a complex wave function (Euler's formula), and the second term can be understood as a Gaussian windowing function that modulates the amplitude of the wave. Note that the width $\sigma$ of the windowing function falls off with frequency.

For a given frequency $f$, the superlet transform of order $o$ is the geometric mean of Morlet transforms $\psi_{c,f}$ for a range of values $c \in [1, o]$. The superlet representation at a frequency of interest $f$ for a signal $x$ is then:

$$\sqrt[o]{\prod_{i=1}^{o} \sqrt{2} \cdot x * \psi_{c,f}}, \tag{6}$$

where $*$ is the complex convolution operator. Although, compared to the Morlet wavelet transform, the overall resolution is improved, the $f$ in the denominator of eq. (5) means that there continues to be relatively lower time resolution at lower frequencies. In other words, there is more temporal smearing at the bottom of the spectrogram than at the top. To account for this, we use an adaptive masking strategy (see section 2: Pretraining).

## B  TIME-FREQUENCY REPRESENTATION PARAMETERS

For the STFT, we use a window size of 400 samples ($\approx 200ms$) with an overlap of 350 samples ($\approx 175ms$) with frequency channels evenly spaced from 0 to 200Hz.

For the superlet, we use $c_1 = 1$ with orders $o = 3-30$. The frequencies of interest are evenly spaced from 0.1 to 200Hz. To match the down-sampling rate of the STFT, the superlet representation is decimated by a factor of 50. Finally, for both representations, we remove 5 columns from each array on either side ($\approx 250ms$) to account for edge-effects.

Superlets take the composite of many Morlet wavelets. Thus, a superlet is simply a collection of Morlet wavlets with different $c$:

$$SL_c = \{\psi_{f,c_i} \mid c_1, ..., c_o\} \tag{7}$$

where $o$ is the order of the superlet.

For this work, we use the multiplicative, adaptive superlet transform. In the multiplicative version of the transform, $c_i = c_1 \cdot i$. And in the adaptive version, the order of the superlet changes according to frequency:

$$o = o_{min} + \left[(o_{max} - o_{min}) \cdot \frac{f - f_{min}}{f_{min} - f_{max}}\right] \tag{8}$$

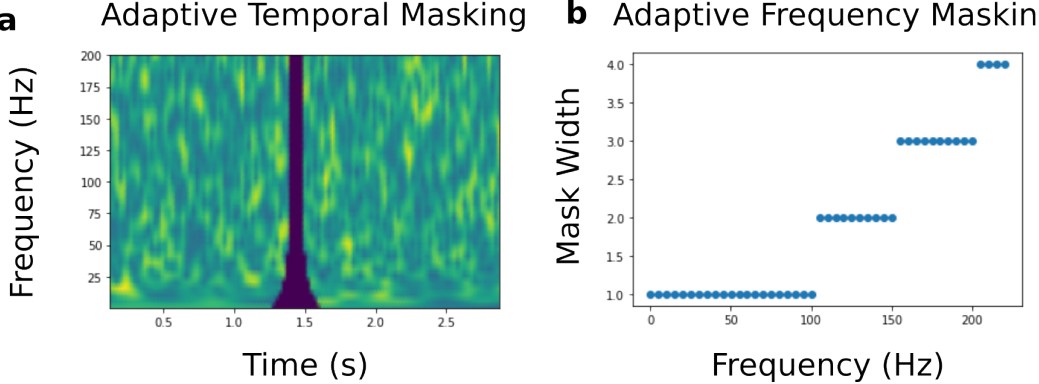

Figure 7: **Adaptive masking** *(Left)* The adaptive temporal mask, for which the width increases with the inverse of frequency. *(Right)* The width of the frequency mask, which specifies the amount of masked frequency channels, increases as a function of frequency.

## C  MASKING TIME AND FREQUENCY

During pre-training, BrainBERT receives an augmented version of the spectrogram, from which random time and frequency bands have been removed. In this work, we use two masking strategies: static masking, which is suited for spectrograms of fixed time-frequency resolution, and adaptive masking, which is appropriate for spectrograms with a variable trade-off in time-frequency resolution. **Static masking**    For our static masking procedure, we adapt the work of (Liu et al., 2020; 2021). The width of the temporal mask is randomly selected from the range $[1, 5]$, and the width of the frequency mask is randomly selected from the range $[1, 2]$.

**Adaptive masking**    For the superlet transform, we use adaptive temporal and frequency masks. The temporal width of the adaptive mask as a function of frequency $f$ is given by:

$$w_t(f) = 2 \max \left( m, \frac{200}{20 + f} \right) \tag{9}$$

where $m$ is the minimum width of the mask, which is randomly selected from $\{1, 2\}$ per example. See fig. 7 for visualization.

The width of the frequency masks is given as a function of frequency $f$:

$$w_f(f) = \max \left( 1, \left\lfloor \frac{4.9 f}{250} \right\rfloor \right) \tag{10}$$

The parameters of these equations are selected to roughly match the width of the masks used for the STFT.

## D  MASKING PROCEDURE

The masking procedure is described in algorithm 1 (see section 2: Pretraining). Note that intervals are masked without overlap. The procedure is adapted from Liu et al. (2021) and Devlin et al. (2019). During pre-training, we use $p_{\text{mask}} = 0.05$, $p_{ID} = 0.1$, and $p_{\text{replace}} = 0.1$. The purpose of letting some segments go unaugmented is to make the distribution seen at train time and test time more similar. The purpose of replacing some segments with random signal is to prevent the model from simply learning the identity function.

---

**Algorithm 1** Time-masking procedure

---

$\mathbf{Y} \leftarrow n \times m$ spectrogram
$i \leftarrow 0$
**while** $i \leq m$ **do**
    $p \sim \text{Unif}(0, 1)$
    **if** $p < p_{\text{mask}}$ **then**
        $l \sim \lfloor \text{Unif}(\text{step}_{\text{min}}, \text{step}_{\text{max}} + 1) \rfloor$
        $q \sim \text{Unif}(0, 1)$
        **if** $q < p_{\text{ID}}$ **then**
            **pass**
        **else if** $p_{\text{ID}} \leq q < p_{\text{ID}} + p_{\text{replace}}$ **then**
            $j \leftarrow \text{Unif}(0, m - l)$
            $\mathbf{Y}[:, i : i + l] \leftarrow \mathbf{Y}[:, j : j + l]$
        **else**
            $\mathbf{Y}[:, i : i + l] \leftarrow \mathbf{0}$
        **end if**
        $i \leftarrow i + l$
    **end if**
**end while**

---

## E    BASELINES

**Linear baselines (time domain)**    We train two feed forward networks with layer dimensions $[d_{\text{input}}, 1]$ and sigmoid activations. The first model, Linear (5s time domain), takes 5s of time domain input, sampled at 2048 Hz, so $d_{\text{input}} = 10, 240$. The second model, Linear (0.25s time domain), takes 0.25s of input, so $d_{\text{input}} = 512$.

**Deep neural network (time domain)**    We train a deep neural network, Deep NN (5s time domain), which consists of 5 stacked feed forward layers with dimensions $[d_{\text{input}}, 1024, 512, 256, 128]$ and ReLU activations on the hidden layers and a sigmoid activation on the output layer. The network takes 5s of time domain input, sampled at a rate of 2048 Hz, so $d_{\text{input}} = 10, 240$.

**Linear baselines (time-frequency domain)**    We train two feed forward linear networks with dimension $[d_{\text{input}}, 1]$ and sigmoid activations that take the time-frequency representations as input. Linear (.25s STFT) and Linear (.25s superlet) receive the time-frequency representation, averaged across time in a $\approx 244$ms interval, centered on the example. More precisely, given a time-frequency representation $\mathbf{Y} \in \mathbb{R}^{n \times 2l}$, the model receives the vector obtained by averaging $Y_{:, l-5, l+5}$ across the time axis. There are $n = 40$ frequency bands, so $d_{\text{input}} = 40$. This is the same form as the input that our classification network receives from BrainBERT (see section 3).

## F    PREPROCESSING

Data was high-pass filtered at 0.1Hz. Line noise at 60Hz and its harmonics were removed. Each electrode was re-referenced by subtracting out the mean signal from the two adjacent electrodes on the same shaft (Laplacian re-referencing). This decreases the cross-correlation between electrodes (Li et al., 2018). Only electrodes which can be Laplacian re-referenced, i.e., have neighbors, are included in the pretraining data. Finally, signals from all electrodes are visually inspected, and recordings which show obvious signs of corruption are removed. In total, data from 1,249 electrodes ($\approx 74\%$ of total electrodes; 4,551 electrode-hours) are used for pretraining.

## G    PRETRAINING PARAMETERS

All layers ($N = 6$) in the encoder stack are set with the following parameters: $d_h = 768$, $H = 12$, and $p_{\text{dropout}} = 0.1$. We pretrain the BrainBERT model with the LAMB optimizer (You et al., 2019) and $lr = 1e - 4$. We use a batch size of $n_{\text{batch}} = 256$, train for 500k steps, and record the validation performance every 1,000 steps. Then, the weights with the best validation performance are retained.

During pretraining, the BrainBERT representations are passed to a spectrogram prediction head which consists of a two feed forward linear layers. The first layer, a hidden layer, has dimension

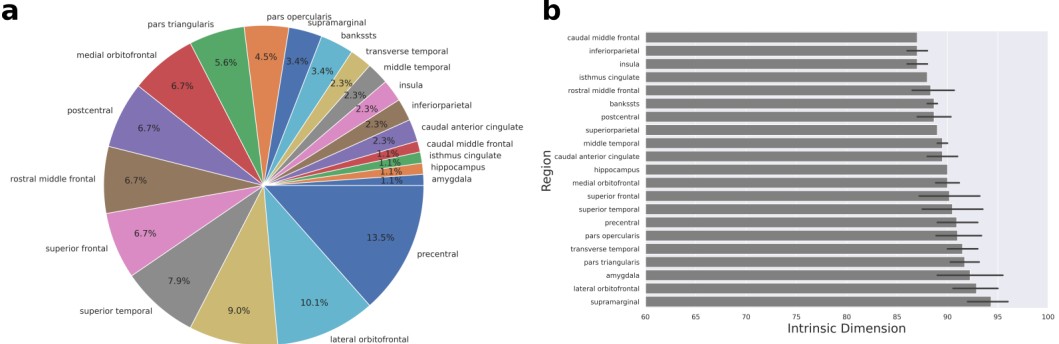

Figure 8: **Intrinsic dimension averaged by region** We find the intrinsic dimension (ID) for all held-out electrodes, and consider the electrodes in the top 10-th percentile. Among these electrodes, the percentage which fall in each region is shown in (a). However, the electrodes are not distributed uniformly across the brain, so to get a normalized view of the regions, we take the mean across electrodes in each region (b). Error bars show a 95% confidence interval

.

$d = 768$ and a GeLU activation. The second layer, the output layer, has dimension $d = 40$ This matches the height of the spectrogram, which is determined by the number of frequencies of interest; see appendix B.

As part of our ablation test, we also include a BrainBERT with randomly initialized weights. For this, weights are initialized according to the uniform Xavier scheme (Glorot & Bengio, 2010).

## H FINE-TUNING TRAINING PARAMETERS

The classifier is a fully connected linear layer with $d_{\text{in}} = 768$, $d_{\text{out}} = 1$ and a sigmoid activation. The model is trained using a binary cross entropy loss. When fine-tuning, we use the AdamW optimizer, with $lr = 1e - 3$ for the classification head and $lr = 1e - 4$ for the BrainBERT weights. When training with frozen BrainBERT weights, we use the AdamW optimizer with $lr = 1e - 3$.

All models, both this classifier and the baseline models, are trained for 1,000 updates. Validation performance is computed every 100 updates, and test accuracy is reported using the weights with the best validation performance.

## I INTRINSIC DIMENSION

We use a standard method for determining intrinsic dimension (Fan et al., 2010). For a given threshold $\beta$, the intrinsic dimension is the $d \in \mathbb{N}$ such that the ratio of explained variance for $d$ dimensions of a $N$ dimensional PCA is above $\beta$:

$$\frac{\sum_{i=1}^{d} var(y_i)}{\sum_{j=1}^{N} var(y_j)} > \beta \tag{11}$$

We use a PCA of $N = 200$ and a threshold of $\beta = 0.95$. For each session in the held out data, we segment the neural recordings into 5s intervals, for which the spectrogram is a matrix $\mathbf{Y} \in \mathbb{R}^{n \times m}$. We produce the BrainBERT embeddings $\mathbf{E} \in \mathbb{R}^{n \times m}$ for each spectrogram, and average across the time-dimension to obtain a single $n$-dimensional vector representation of the interval. For each electrode, we find the intrinsic dimension of the manifold on which these BrainBERT embeddings lie. We find the regions with the highest intrinsic dimension, by first taking the electrodes in the top 10-th percentile. Among these electrodes, most of them lie in the precentral gyrus (13.5%), the lateral orbitofrontal cortex (10.1%), and the pars opercularis (9.0%). However, it should be noted that the electrodes are not distributed evenly across the cortex, so to get a sense of which regions have the highest ID, we should normalize across the number of electrodes in a region. We find that

the regions with the highest mean ID are the supramarginal gyrus, the lateral orbitofrontal cortex, and the amygdala.

## I.1 TASKS

**Sentence onset**    Once the movie transcript has been aligned to the brain activity, the brain activity which corresponds to the sentence onsets can be collected, each embedded in 5s of context. These intervals form the set of positive examples.

**Speech vs. non-speech**    For this task, the positive examples are formed from the 5s of context surrounding any word. For both this task and the sentence onset task, the negative examples are formed by finding 1s intervals which do not overlap with any intervals of speech audio. Each example is embedded in 5s of context.

**Volume**    For each word in the transcript, the volume is measured as the root-mean-square for a 500ms interval starting from the word onset. Those words which lie one standard deviation above the mean volume are labeled as "high-volume". Those words which lie one standard deviation below the mean volume are labeled as "low-volume".

**Pitch**    For each word in the transcript, the pitch is extracted using librosa's `piptrack` function over a Mel-spectrogram (sampling rate 48,000 Hz, FFT window length of 2048, hop length of 512, and 128 mel filters). Those words which lie one standard deviation above the mean pitch are labeled as "high-pitch". Those words which lie one standard deviation below the mean pitch are labeled as "low-vpitch".

## J  DATASET STATISTICS

| Subject | Sentence onset | | Speech/Non-speech | | Volume | | Pitch | |
|---|---|---|---|---|---|---|---|---|
| | Onset | Non-speech | Speech | Non-speech | High | Low | High | Low |
| subject-1 | 1,587 | 1,587 | 6,333 | 6,333 | 1,135 | 542 | 1,723 | 1,724 |
| subject-2 | 1,071 | 1,071 | 6,413 | 6,413 | 662 | 137 | 1,042 | 1,066 |
| subject-3 | 2,057 | 2,057 | 2,581 | 2,581 | 1,350 | 980 | 1,591 | 1,592 |
| subject-4 | 542 | 542 | 2,500 | 2,500 | 295 | 229 | 378 | 364 |
| subject-5 | 1,059 | 1,059 | 3,183 | 3,183 | 769 | 521 | 1,009 | 980 |
| subject-6 | 1,668 | 1,668 | 2,367 | 2,367 | 1,092 | 815 | 1,311 | 1,309 |
| subject-10 | 1,971 | 1,971 | 1,971 | 1,971 | 1,350 | 980 | 1,591 | 1,592 |

Table 3: **Annotated data statistics** The number of examples per task and per subject for the held-out sessions are shown here. For the sentence onset task and speech vs. non-speech task, the number of examples is explicitly balanced between classes. Non-speech examples correspond with 1s intervals which do not overlap with any word-audio in the movie.

| Subj. | Age (yrs.) | # Electrodes | Movie | Recording time (hrs) | Held-out |
|---|---|---|---|---|---|
| 1 | 19 | 91 | Thor: Ragnarok | 1.83 | |
| | | | Fantastic Mr. Fox | 1.75 | |
| | | | The Martian | 0.5 | x |
| 2 | 12 | 100 | Venom | 2.42 | |
| | | | Spider-Man: Homecoming | 2.42 | |
| | | | Guardians of the Galaxy | 2.5 | |
| | | | Guardians of the Galaxy 2 | 3 | |
| | | | Avengers: Infinity War | 4.33 | |
| | | | Black Panther | 1.75 | |
| | | | Aquaman | 3.42 | x |
| 3 | 18 | 91 | Cars 2 | 1.92 | x |
| | | | Lord of the Rings 1 | 2.67 | |
| | | | Lord of the Rings 2 (extended edition) | 3.92 | |
| 4 | 9 | 135 | Megamind | 2.58 | |
| | | | Toy Story | 1.33 | |
| | | | Coraline | 1.83 | x |
| 5 | 11 | 205 | Cars 2 | 1.75 | x |
| | | | Megamind | 1.77 | |
| 6 | 12 | 152 | Incredibles | 1.15 | |
| | | | Shrek 3 | 1.68 | x |
| | | | Megamind | 2.43 | |
| 7 | 6 | 109 | Fantastic Mr. Fox | 1.5 | |
| 8 | 4.5 | 102 | Ant Man | 2.28 | |
| 9 | 16 | 72 | Sesame Street Episode | 1.28 | |
| 10 | 12 | 173 | Cars 2 | 1.58 | x |
| | | | Spider-Man: Far from Home | 2.17 | |

Table 4: **Subject statistics** Subjects used in BrainBERT training, and held-out downstream evaluation. The number of uncorrupted, electrodes that can be Laplacian re-referenced are shown in the second column The average amount of recording data per subject is 4.3 (hrs).

## K    ELECTRODE VISUALIZATION

For each subject, pre-operative T1 MRI scans without contrast were processed with FreeSurfer (Fischl et al., 2004), which performed skull stripping, white matter segmentation, and surface generation. iELVis (Groppe et al., 2017) was used to co-register a post-operative fluoroscopy scan to the preoperative MRI. Electrodes were manually identified using BioImageSuite (Joshi et al., 2011), and then assigned to one of 94 regions (according to the Desikan-Killiany atlas (Desikan et al., 2006)) using FreeSurfer's automatic parcellation. The alignment to the atlas was manually verified for each subject.

Depth electrodes in the white matter, if they were within 1.5 mm of the gray-white matter boundary, were projected to the nearest point on that boundary, and were labeled as coming from that region (for the purposes of region analyses). This procedure is very similar to the post brain-shift correction methods used for electrocorticography electrodes (Yang et al., 2012). For solely visualization purposes, all electrodes identified to lie in the gray matter or on the gray-white matter boundary were first projected to the pial surface (using nearest neighbors), and then mapped to an average brain (using Freesurfer's fsaverage atlas) for the visualizations shown in fig. 3 and fig. 6.

## L    PRETRAINING PERFORMANCE

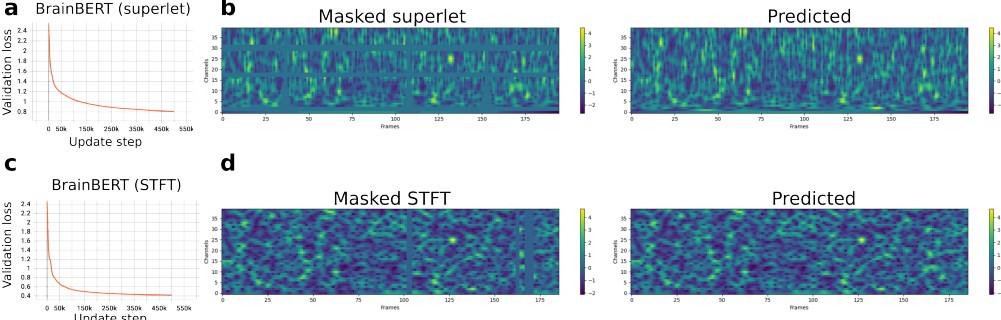

Figure 9: **Pretraining performance** (a) During pretraining, BrainBERT with superlet inputs achieves 0.81 content-aware reconstruction loss. The L1 reconstruction component (not pictured) of this loss is 0.41. (c) BrainBERT with STFT inputs achieves 0.42 content aware reconstruction loss, of which the L1 component accounts for 0.20. (b) and (d) show sample reconstructions produced by BrainBERT after 500k updates.

# M  Supplementary Figures

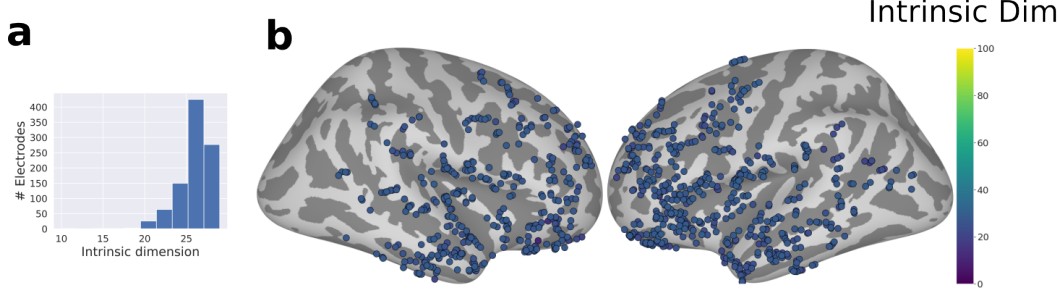

Figure 10: **Intrinsic dimension of STFT features** In fig. 6, we show plot the intrinsic dimension for each electrode based on the BrainBERT representation of that electrode's activity. For comparison, a similar plot (b) can be made using the raw short-time Fourier transform (STFT) features. Unlike for the BrainBERT representations, the distribution of dimensions is tightly grouped around a few values (a).

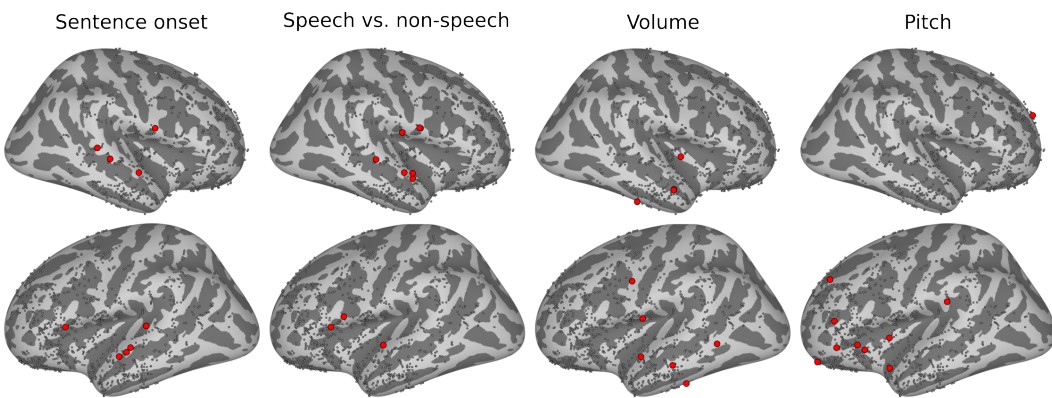

Figure 11: **Location of test electrodes** The results in section 4 are given according to a fixed subset of electrodes per task. For each task, this subset is determined by training a linear classifier over all electrodes and taking the top 10 electrodes with the best performance. The location of these electrodes is shown here.

## N    SUPPLEMENTARY RESULTS

|        |                            | Sent. onset | Speech/Non-speech | Pitch | Volume |
|--------|----------------------------|-------------|-------------------|-------------|-------------|
| Top 10 | Linear (5s, time domain)   | .63±.04     | .58 ± .06         | .58 ± .07   | .56 ± .19   |
|        | BrainBERT (STFT)           | .82±.07     | .93 ± .03         | .75 ± .03   | .83 ± .09   |
|        | BrainBERT (superlet)       | .78±.08     | .86 ± .06         | .62 ± .05   | .70 ± .10   |
| Top 20 | Linear (5s, time domain)   | .63±.04     | .57 ± .04         | .58 ± .06   | .61 ± .10   |
|        | CortexBERT (STFT)          | .82±.08     | .91 ± .10         | .74 ± .06   | .85 ± .07   |
|        | CortexBERT (superlet)      | .77±.10     | .81 ± .12         | .68 ± .06   | .76 ± .09   |

Table 5: Performance is evaluated per task and per electrode. In order to keep computational costs reasonable, since there are many baselines and ablations to compare against, we only evaluate the performance for a subset of all electrodes. For a given task, this subset is selected by first training a linear classifier over all electrodes. In section 4, we find the top 10 electrodes with the highest linear classifier performance. This subset is then held fixed over all comparisons. The variance reported is calculated with respect to this set of electrodes. In this table, we report the results for the top 20 electrodes as well for comparison.

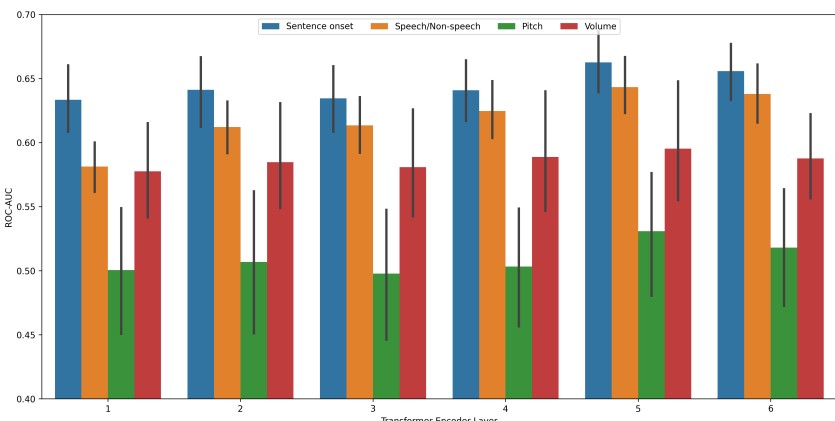

Figure 12: **Layerwise analysis** We compute the decoding performance of the frozen BrainBERT features per layer of the transformer encoder stack. We see that for all tasks, performance increases with depth in the stack, peaking at the second to last layer. Note that in this work, we used the features taken from the last layer. And for the purposes of showing the advantages of BrainBERT's pre-training, the performance we obtain is sufficient.

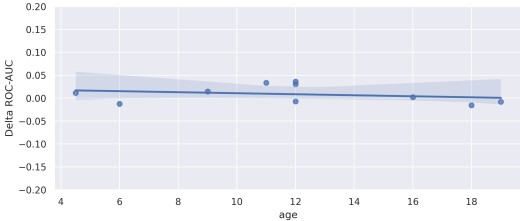

Figure 13: **Generalization vs. age.** The ability of BrainBERT to transfer to new subjects does not depend on age (Pearson's $r = -0.27$, with $p$-value $0.451$). The y-axis shows the delta between the AUC-ROC, averaged across all tasks, between BrainBERT pretrained with and without a particular subject. The x-axis shows the age of the subject. See table 6 for complete breakdown per task.

| | Sentence onset | Speech/Non-speech | Pitch | Volume | Average |
|---|---|---|---|---|---|
| pretrain w/ subject 1 | .74±.09 | .85 ± .06 | .79 ± .05 | .92 ± .03 | .82 ± .09 |
| pretrain w/o subject 1 | .73±.09 | .85 ± .05 | .77 ± .10 | .92 ± .03 | .82 ± .10 |
| Linear (5s) | .62±.04 | .56 ± .04 | .52 ± .02 | .62 ± .06 | .58 ± .06 |
| pretrain w/ subject 2 | .61±.05 | .83 ± .09 | .74 ± .03 | .73 ± .10 | .73 ± .11 |
| pretrain w/o subject 2 | .62±.08 | .85 ± .07 | .77 ± .03 | .82 ± .13 | .77 ± .12 |
| Linear (5s) | .59±.06 | .53 ± .03 | .52 ± .05 | .61 ± .11 | .56 ± .08 |
| pretrain w/ subject 3 | .85±.07 | .81 ± .10 | .65 ± .03 | .86 ± .10 | .79 ± .12 |
| pretrain w/o subject 3 | .84±.05 | .78 ± .14 | .61 ± .07 | .88 ± .03 | .78 ± .13 |
| Linear (5s) | .64±.02 | .57 ± .03 | .54 ± .04 | .56 ± .03 | .58 ± .05 |
| pretrain w/ subject 4 | .57±.07 | .94 ± .03 | .66 ± .08 | .69 ± .06 | .72 ± .15 |
| pretrain w/o subject 4 | .60±.08 | .91 ± .12 | .66 ± .06 | .75 ± .09 | .73 ± .15 |
| Linear (5s) | .58±.05 | .54 ± .02 | .61 ± .04 | .64 ± .05 | .59 ± .05 |
| pretrain w/ subject 6 | .63±.11 | .76 ± .12 | .69 ± .03 | .70 ± .06 | .70 ± .10 |
| pretrain w/o subject 6 | .64±.11 | .73 ± .16 | .76 ± .03 | .78 ± .04 | .73 ± .11 |
| Linear (5s) | .56±.04 | .53 ± .04 | .53 ± .06 | .58 ± .06 | .55 ± .06 |
| pretrain w/ subject 7 | .74±.06 | .91 ± .04 | .68 ± .03 | .83 ± .08 | .79 ± .10 |
| pretrain w/o subject 7 | .75±.06 | .86 ± .13 | .66 ± .06 | .85 ± .02 | .78 ± .11 |
| Linear (5s) | .63±.04 | .57 ± .05 | .58 ± .03 | .53 ± .05 | .58 ± .05 |
| pretrain w/ subject 5 | .71±.09 | .75 ± .02 | .65 ± .03 | .84 ± .04 | .74 ± .09 |
| pretrain w/o subject 5 | .74±.05 | .81 ± .04 | .65 ± .07 | .89 ± .03 | .77 ± .10 |
| Linear (5s) | .56±.05 | .53 ± .04 | .52 ± .04 | .55 ± .03 | .54 ± .04 |
| pretrain w/ subject 8 | .68±.10 | .89 ± .13 | .66 ± .05 | .76 ± .04 | .74 ± .12 |
| pretrain w/o subject 8 | .70±.06 | .85 ± .12 | .72 ± .06 | .75 ± .08 | .76 ± .10 |
| Linear (5s) | .61±.04 | .54 ± .04 | .59 ± .04 | .57 ± .05 | .58 ± .05 |
| pretrain w/ subject 9 | .75±.11 | .80 ± .09 | .75 ± .04 | .59 ± .21 | .72 ± .15 |
| pretrain w/o subject 9 | .78±.06 | .80 ± .10 | .76 ± .03 | .56 ± .16 | .72 ± .14 |
| Linear (5s) | .55±.04 | .53 ± .02 | .53 ± .02 | .80 ± .08 | .60 ± .12 |
| pretrain w/ subject 10 | .79±.05 | .81 ± .12 | .68 ± .07 | .90 ± .05 | .80 ± .11 |
| pretrain w/o subject 10 | .76±.06 | .84 ± .10 | .70 ± .03 | .86 ± .08 | .79 ± .09 |
| Linear (5s) | .62±.04 | .53 ± .03 | .53 ± .04 | .56 ± .05 | .56 ± .06 |

Table 6: **Held-one-out analysis**. For each subject, we report the performance of two versions of BrainBERT with superlet input. One version is trained without that particular subject, and one version is trained using all subjects. Fine-tuning performance is reported on a subset of electrodes belonging to the subject in question. As before, this subset is determined by running a linear classifier over all electrodes and taking the top 10 electrodes with the highest performance. The performance of that linear classifier is also shown for comparison. In general, we see a close match between the two versions of BrainBERT. In some instances, the version without a particular subject even performs better. This can be attributed to the fact that one subject's data may have a negative contribution when it comes to learning useful representations for a particular task.

