# OpenReview forum: "BrainBERT: Self-supervised representation learning for intracranial recordings"
_ICLR.cc/2023/Conference — ICLR 2023 poster_

### Official Review · Reviewer_1MuJ · 2022-10-24

**Confidence:** 4
**Correctness:** 3
**Technical Novelty And Significance:** 3
**Empirical Novelty And Significance:** 1
**Recommendation:** 6

**Clarity, Quality, Novelty And Reproducibility:**

The clarity, quality, and reproducibility of the work are solid.
The novelty is also good within the neuroscience community, though the approach heavily builds off of existing approaches for speech self-supervised models.




**Details Of Ethics Concerns:**

The proposed approach integrates brain recordings from individuals into a pretrained model that will be made publicly available. The authors need to discuss any possible implications for privacy and security.

**Strength And Weaknesses:**

Strengths:
- clearly written for the most part
- potentially impactful (if major concerns are addressed, see below)
- timely

Weaknesses:
Major:
1. the 10 subjects used to pretrain the model are very young (ages 4-19, the distribution of ages among the subjects is not clear) and it's not clear how well data from these subjects would generalize to a more general population. The work shows some results from holding out one of these subjects, but it's not clear whether these results are averaged over holding out each of the subjects, or they are for only one randomly selected subject. It would be helpful if the authors can clarify this and show whether there are any differences in results when generalizing to subjects of different ages. The authors should also discuss this possible limitation of their pretrained model.
2. a truly general-purpose pretrained model would be able to generate brain activity affected by different stimuli and tasks, but there is no validation of this ability of the model in the current work. How do the authors envision this model be used -- with intracranial recordings collected under any stimulus and any task, or only for naturalistic movie stimuli with passive viewing? It's important that this is discussed in the work.
3. the tested "tasks" for which the pretrained embeddings are useful are claimed to vary from low-level to high-level, but to me all of these tasks are indeed low-level because they should not require any processing beyond early auditory cortex. It would be more convincing if a higher-level task, such as word-level decoding, is enabled by the pretrained model

Minor:
- the name BrainBERT would be more fitting for a model that incorporates multiple types of neural recordings. A name that more accurately reflects the actual recordings used would make it easier to remember and distinguish from any follow-up work that incorporates other types of recordings.

Clarifications / Questions:
1. some of the baseline models are said to have "randomized weights" -- does this actually mean that the weights were randomized or that they were randomly initialized? Randomly initialized weights would be more convincing as a baseline, since it's not clear what the weight distribution is and how effective randomizing them would be for reducing the effect of training
2. how was the stopping point for training determined for all models?
3. are the intrinsic dimensionality results only possible because of the pretrained model? Can these analyses be done directly with the raw data or the Fourier transform, and if so, would the results be different?

**Summary Of The Paper:**

I'm increasing my score from 5 to 6 as the authors have addressed several of my concerns.
However, I am still not convinced about the utility of this architecture or pretrained weights beyond the specific experimental setting that is considered by the authors because the task someone is performing has a big effect on the brain activity. I would expect a paper that claims to deliver the BERT of brain recordings to show this. So I really hope that the authors indeed include a disclaimer about this in their manuscript.

^^^^^^^^^^^ POST-REBUTTAL ^^^^^^^^^^^^^^^^^^

This work proposes an approach to pretrain a transformer from scratch, akin to BERT (though with fewer layers), to predict masked intracranial recordings of people viewing movies. The pretrained model can generalize to heldout subjects and electrode positions. The claim /hope is that this pretrained model can be used, similarly to pretrained models in NLP, as a multipurpose tool that can be easily fine-tuned for "downstream" neuroscience tasks. The representations of intracranial recordings that are constructed by the pretrained BrainBERT lead to higher decoding accuracy in several "tasks" (e.g predicting the volume of the stimulus audio, predicting whether the stimulus is speech vs non-speech, etc), than other baseline representation methods (e.g. the raw recordings, a Fourier transform of the raw recordings, a 5 layer feed forward neural network).




**Summary Of The Review:**

In theory, this work should represent an advance for the computational neuroscience community, but it is not clear how general-purpose the representations that this model produces will be to brain recordings from different types of stimuli and different types of experimental tasks. I would change my assessment of the work, if the authors are able to address the three major concerns that I described above.

---

> ### Author Response · Authors · 2022-11-16
> **Response to reviewer suggestions**
>
> Thank you for the helpful suggestions and questions. The reviewer’s suggestions have prompted us to revise our analyses, especially with respect to generalizability and intrinsic dimension. Below, we provide a point-by-point response.
>
> # Major topics
> > it's not clear how well data from these subjects would generalize to a more general population. … It would be helpful if the authors can clarify this and show whether there are any differences in results when generalizing to subjects of different ages.
> - The ages of all subjects are now shown in “Appendix J: Dataset statistics”. There is no relationship between generalization performance to new subjects and the age of that subject (see below answer, Figure 13 of the revised text). Of course, this might be different if we had much older subjects, but in that case we would include a few such subjects into the pretraining set.
>
> > The work shows some results from holding out one of these subjects, but it's not clear whether these results are averaged over holding out each of the subjects, or they are for only one randomly selected subject.
> - This is fair. We expanded our analysis, to perform a full hold-one-out analysis across all subjects. The results, averaged across tasks, can be seen in Figure 3. The complete results per task can be seen in “Appendix M: Supplementary Results.” The cross-validated results are essentially the same as our earlier results with only a single subject, showing that the method generalizes well.
>
> > The authors should also discuss this possible limitation of their pretrained model.
> - Thanks! We now do so in the “Ethics” section. We want to emphasize the difference between the pretrained weights we release and the methodology we are proposing. The pretrained weights will be specific to a particular dataset, but this will be true of any set of weights. The methodology can be used generically for data collected under any stimuli; see answer further below.
>
> > a truly general-purpose pretrained model would be able to generate brain activity affected by different stimuli and tasks, but there is no validation of this ability of the model in the current work.
> - Regarding generation -- this is an interesting point! Only as of the past few months have techniques emerged that can reliably generate data from Transformers, with the advent of diffusion models. It would be interesting to see if we could do so from BrainBERT. We will look into this for a followup and perhaps for new applications. Thanks!
>
> > How do the authors envision this model be used -- with intracranial recordings collected under any stimulus and any task, or only for naturalistic movie stimuli with passive viewing? It's important that this is discussed in the work.
> - We will add this to the manuscript. We note that nothing in our method is specific to passive movie viewing. We restricted the models we trained so far to that data to make evaluation practical for the publication, but we don’t use the stimuli themselves to inform our model architecture or objective. Only the neural recordings are used during pretraining, in a task-and-stimuls agnostic way. Indeed, we will release a larger variant of the model down the road trained at scale with full-day recordings using the same methodology.
>
> > the tested "tasks" for which the pretrained embeddings are useful are claimed to vary from low-level to high-level, but to me all of these tasks are indeed low-level because they should not require any processing beyond early auditory cortex. It would be more convincing if a higher-level task, such as word-level decoding, is enabled by the pretrained model
> - Determining sentence onset [1] and distinguishing speech vs. non speech [2,3] are tasks that engage the language processing regions of the brain. The reviewer is right in that some of the tasks, such as determining whether or not a noise is loud can be solved by early auditory cortex. This is why we refer to a range of low to high level tasks.
>
> # References
> [1] Fedorenko, Evelina, et al. "Neural correlate of the construction of sentence meaning." Proceedings of the National Academy of Sciences (2016)
>
> [2] Möttönen, Riikka, et al. "Perceiving identical sounds as speech or non-speech modulates activity in the left posterior superior temporal sulcus." Neuroimage (2006)
>
> [3] Parviainen, Tiina, Päivi Helenius, and Riitta Salmelin. "Cortical differentiation of speech and nonspeech sounds at 100 ms: implications for dyslexia." Cerebral Cortex (2005)

---

> > ### Comment · Reviewer_1MuJ · 2022-11-22
> > **more clarification needed**
> >
> > I thank the reviewers for taking the feedback seriously and providing some additional analyses regarding the generalization to held-out subjects. There is still one major remaining concern that I believe was misunderstood from the original review:
> >
> > > a truly general-purpose pretrained model would be able to generate brain activity affected by different stimuli and tasks, but there is no validation of this ability of the model in the current work.
> > >>Regarding generation -- this is an interesting point! Only as of the past few months have techniques emerged that can reliably generate data from Transformers, with the advent of diffusion models. It would be interesting to see if we could do so from BrainBERT. We will look into this for a followup and perhaps for new applications. Thanks!
> >
> > By generate brain activity, I meant exactly what your model is trained to do -- taking as input masked time-freq representations of brain activity and predicting as output the masked parts of the time-freq representation. The question is, do you expect that you can use your pretrained model to faithfully output masked parts of time-freq representations of brain recordings that are obtained from humans viewing other types of stimuli and for doing other types of tasks than passive viewing. This question is related to the next question about how the authors envision their model be used. The true general utility of your model depends on answering this question, and I do not see any attempts currently to answer it.
> >
> > And I also have a few more minor points that I do not consider deal breakers, unlike the concern above.
> > 1. New results on generalization to held-out subjects in appendix N (Fig 13 and Table 6): my understanding is that these results are obtained after a model is fine-tuned with some of the held-out subject's data. That is ok (as long as of course the final evaluation is done on a completely heldout set of data from this "held-out" subject). However, the fine-tuning makes it more difficult to determine whether the pretrained weights are the reason your model is generalizing so well to data from pretraining-heldout subjects. Alternatively, it may be the architecture + fine-tuning that is leading to the good performance at the end. Therefore, another baseline which would be useful to compare against is fine-tuning a randomly initialized model in the same way that you fine-tune the pretrained one and evaluating on the same held-out data. This will likely lead to results that are on par with the results shown in Table 1 which, as I understand, are for fine-tuning a randomly initialized model without holding out any subject. But it would still be more complete to actually calculate these baselines for each of the heldout subjects.
> >
> > 2. The limitation regarding the type of population and data that was used to pretrain the model should be discussed in the main paper under discussion or conclusions/future work, and not only under ethics. The type of problems that arise from the pretraining on particular population are different for the purpose of these different sections: the discussion section should discuss any problems for the performance & generalization that may arise from the type of pretraining data, and the Ethics section should discuss any potential negative impact of releasing a pretrained model on brain data (for example, could someone use this model to identify some of the participating subjects in the pretraining?)

---

> > > ### Author Response · Authors · 2022-11-23
> > > **Clarification follow up**
> > >
> > > # Major points
> > >
> > > Thank you for clarifying! For our evaluation, we use brain activity that was induced by passive viewing, but the type of stimulus is not built into the BrainBERT architecture, pretraining objective function, or masking strategy. And there is no reason why the BrainBERT pretraining approach cannot be used for generic brain activity.
> > >
> > > Whether the *pretrained* *weights* that we release are the best off-the-shelf representations for all possible types of brain activity is an open question, and is unlikely to be true for every stimulus. But we have shown that the pretrained weights are beneficial across a variety of subjects, electrode locations, and classification tasks, and this suggests that they will at least confer an advantage over raw data input. Indeed if one has a large amount of data induced by a significantly different stimulus, the recommended course of action would be to try to use our pretrained weights first and then to pretrain BrainBERT from scratch on the new data if additional performance is needed.
> > >
> > > Here, we draw a comparison to the original BERT model, which was trained mainly on English Wikipedia articles, but has since found a variety of important NLP applications for texts that fall well outside the genre of encyclopedia articles. Subsequent researchers made adjustments to BERT when applying it to different languages and domains, often pretraining it from scratch, but those contributions were separate and came after the initial research effort of developing the BERT method.
> > >
> > > Thank you for raising these questions. This is an important point of discussion to clarify, and we will modify our discussion section in the text to reflect the limitations and recommended use cases of our off-the-shelf weights. In particular, we’ll move these comments out of the ethics section and into the main text, as you suggest.
> > >
> > >
> > > # Minor points
> > > > another baseline which would be useful to compare against is fine-tuning a randomly initialized model in the same way that you fine-tune the pretrained one and evaluating on the same held-out data. This will likely lead to results that are on par with the results shown in Table 1 which, as I understand, are for fine-tuning a randomly initialized model without holding out any subject. But it would still be more complete to actually calculate these baselines for each of the heldout subjects.
> > >
> > > If the reviewer does not feel strongly about this point, then we hope they understand if we do not include this analysis, mainly for the sake of organization and clarity of exposition. Indeed we could have recapitulated the entire comparison between ablations, baselines, and BrainBERT variants per held out subject, but this would have decreased the legibility of the main generalization findings.
> > >
> > > As it stands, the answer to the reviewer’s question about fine-tuning a randomly initialized model can be found in “Section 4: Ablations” and Table 1. To summarize: we fine-tuned a model starting from randomly initialized weights and tested it on held out data and found that performance suffers drastically, showing that architecture + fine-tuning is not sufficient to obtain good performance,
> > >
> > > We thank the reviewer for their thoroughness and engagement in discussion. Taken together, their questions have prompted us to revise and clarify our analyses and discussion, especially regarding generalization and future use cases.

---

> > > > ### Comment · Reviewer_1MuJ · 2022-12-08
> > > > **thanks**
> > > >
> > > > I am still not convinced about the utility of this architecture or pretrained weights beyond the specific experimental setting that is considered by the authors because the task someone is performing has a big effect on the brain activity. I would expect a paper that claims to deliver the BERT of brain recordings to show this. So I really hope that the authors indeed include a disclaimer about this in their manuscript.
> > > >
> > > > However even if the pretraining and architecture are only useful for naturalistic viewing, this work will likely be helpful to the increasing number of scientists who study such data. I also appreciate that the authors responded to the rest of my points. For these reasons, I will increase my score and recommend acceptance.

---

> > > ### Author Response · Authors · 2022-12-05
> > > **Follow up**
> > >
> > > The end of the reviewing period is approaching, and we wanted to check if you had a chance to read our response below. If you have any further questions, please don't hesitate to let us know!

---

> ### Author Response · Authors · 2022-11-16
> **Response to reviewer: minor questions**
>
> # Minor topics and questions
> > some of the baseline models are said to have "randomized weights" -- does this actually mean that the weights were randomized or that they were randomly initialized?
> - The weights are indeed *randomly* *initialized*. We've updated the text and captions in section 4 and appendix G to reflect this.
>
> > how was the stopping point for training determined for all models?
> - During pretraining, BrainBERT was trained for 500,000 updates. Performance on the validation set was computed every 1,000 updates, and the checkpoint with the best validation performance was retained. In the current version, this is discussed in "Appendix G: Pretraining parameters". During finetuning, the classification head + BrainBERT were trained for 1,000 updates. Checkpointing and validation performance was computed every 100 updates, and the weights with the best validation performance was retained. For completeness, we’ve updated the text to include these details in "Appendix H: Fine-tuning parameters".
>
> > are the intrinsic dimensionality results only possible because of the pretrained model? Can these analyses be done directly with the raw data or the Fourier transform, and if so, would the results be different?
> - Good question and we should have done this from the beginning. We now show the plot of intrinsic dimension per electrode over the raw short-time Fourier transform (STFT) features in “Appendix M: Supplementary figures”. There is little variation in intrinsic dimensionality across brain regions with the raw time or frequency domain data (the vast majority of the brain has an ID of 26 or 27). BrainBERT reveals a much richer picture.

---

### Official Review · Reviewer_x1NZ · 2022-10-25

**Confidence:** 4
**Correctness:** 4
**Technical Novelty And Significance:** 3
**Empirical Novelty And Significance:** 1
**Recommendation:** 8

**Clarity, Quality, Novelty And Reproducibility:**

Clarity:
- It is difficult to compare performance on the decoding tasks across some of the manipulations.
    - The averages are reported in the text, but it's not clear what the authors are averaging over since this is not reported in the table. Please include the averages in the tables as well.
    - Authors also report some variance in performance (e.g. AUC +/- value), but unclear what this represents -- e.g. standard deviation over 3 random seeds? Please clarify.

Quality: The model and analyses appear to be of sound quality, with the exception of the subject generalizability claim. Note I did not review the code in detail.

Novelty/originality: The originality here is mainly in the application of existing techniques to neuroscience data, and then making the connection to neural decoding approaches. This is novel enough to warrant a publication at ICLR. The actual decoding results do not present any additional novelty beyond understanding parts of the model itself.

Reproducibility: the authors have included code and plan to publicly release the data. Again, I did not review the code in detail but it appears to be enough to reproduce the analyses once the data is released.

**Strength And Weaknesses:**

Strengths
- Method is novel and generalizes to a held-out subject with different electrode positions
- Authors test a suite of 4 decoding tasks, ranging from low-level (volume/pitch) to high-level feature learning
- Several analyses are provided to understand the contributions of different pipeline choices -- e.g. STFT vs superlet-generated spectrogram, linear decoders with different inputs, a deep NN decoder, fine-tuning the whole network vs. the classification head only. This provides key information for future experimenters that may wish to tune the pipeline for specific tasks, or create a variant of the current proposal.
- Authors promise to make the data and trained models available, which should enable their vision of reusability for neuroscientists
- Method has potential to simplify the development of brain-computer interfaces (BCI)

Weaknesses
- The generalizability claim is quite bold, and ought to be backed up with more data. The authors only show one held-out subject, but then claim that the method generalizes across *subjects* (plural). I would appreciate a complete leave-one-out analysis, leaving out each subject in turn and quantifying the change in classification accuracy with & without subject N.
    - This claim is also made when using the top 10 electrodes across all subjects, and the top 10 electrodes for the held-out subject. To support the generalizability claim across electrodes, it would be useful to see where these electrodes are, or at least see a metric of how far apart they are in cortex. Does the model still generalize if the electrodes are in quite different locations?
- The "few-shot learning" claim is also too strong -- the model does require fewer training samples, and may be within the range of other few-shot learning claims in LM papers (which are typically some fraction of the original training set and represent many labeled samples, rather than computer vision papers which are just 2-5 samples per class). However, it definitely does not fall into the range the authors suggest is crucial for many clinical and BCI studies where, in the authors' words, "subjects may participate in only a few dozen trials"
- The intrinsic dimensionality (ID) analysis is of questionable value as it is presented, and not well-supported by the data as far as I can tell.
    - The authors suggest that there is a clustering of high ID electrodes in motor areas. This does not appear to be true to me -- no more than, say, left lateral PFC, which is where I see a cluster -- perhaps the authors can circle the area they are referring to in Fig. 5.
    - The authors report two areas with the highest ID in the text, with no interpretation other than it is "a novel view of functional regions in a task agnostic way." First of all, it's not clear what the novelty is -- typically neuroscientists report areas with lower and lower ID, and draw conclusions from this. What is the interpretation for a high ID area, and what is unexpected about these two regions having a high ID? Second, this is not actually task agnostic. All the data used to train their model is from movie viewing, and they may have seen extremely different ID numbers for other tasks (e.g. motion tasks that are not just passive perception). I grant that BrainBERT has the potential to be task-agnostic, but it isn't currently.
- The authors suggest that "embeddings of the neural data provide a new means by which to investigate the brain", but they fall short of illustrating how a neuroscientist would do this. What is a conclusion one could draw from the type of data presented here? One might claim that pitch is processed in lower areas of the language processing stream, e.g. auditory cortex. This claim would need to be supported by showing differences in BrainBERT performance when using electrodes from this area vs. electrodes from another area. Even so, a neuroscientist might ask why one should use BrainBERT embeddings instead of training decoders directly on neural data. I encourage the authors to think about what additional utility / inferential power BrainBERT would provide that other methods cannot. Other authors have claimed that using NN models of the brain provide the ability to perform causal interventions (on the model, instead of the brain), which cannot be done with noninvasive neuroimaging methods like fMRI. However, causal interventions can actually be done with iEEG experiments -- so what else might BrainBERT provide?
- Some clarity issues -- see below.

Further questions/suggestions
- It seems the authors are only using the last layer output of the frozen BrainBERT to train the decoders. Is this correct? If so, I'm curious how different layer outputs would change the performance across tasks. It is possible that tasks requiring higher-level features do better when using late layer embeddings, while tasks requiring lower-level features (e.g. pitch/volume) do better when using early layer embeddings.

**Summary Of The Paper:**

EDIT: I have upgraded my overall score (6 to 8), as well as my correctness score (2 to 4).

This paper proposes the use of the bidirectional transformer BERT to model intracranial EEG (iEEG) recordings collected from human patients while watching movie stimuli. They transform the iEEG data into spectrograms and use an approach from speech audio modeling to train the model to reconstruct masked parts of the spectrogram (akin to a masked language modeling objective). They call this proposed model BrainBERT. They then show (1) that the representations learned by BrainBERT can be used for classification tasks, such as speech vs. non-speech, (2) that the method generalizes to a held-out subject with different electrode locations, and (3) a variety of controls and ablation studies providing insight into what parts of the proposal are useful. They assert that this model provides useful embeddings for intracranial recordings in a reusable, subject- and electrode-agnostic way. They also assert that the improved performance on neural decoding tasks is an important contribution to neuroscience, where "many core results in neuroscience hinge on whether a linear decoder can perform a certain task...also critical to building the next generation of brain-machine interfaces."

**Summary Of The Review:**

Overall, I found the proposal and results interesting, and it is certainly a novel application of language modeling approaches that could be quite useful to neuroscience. However, the paper makes many strong claims which are often not supported by the data presented in the paper. I would mainly like the authors to pull back on the strength of some of their claims, and have suggested additional analyses that would bolster some of their current ones. I also encourage the authors to either isolate their claims to BrainBERT's potential use in BCI, or add more content elaborating how a neuroscientist might use the model to better understand the brain. I am rating this a marginal accept because I believe there is some merit to this approach, but the paper needs to be improved. Discussion with the other reviewers and with the authors may change this rating in either direction.

---

> ### Author Response · Authors · 2022-11-16
> **Response to reviewer suggestions**
>
> Thank you for the thorough review. The reviewer’s suggestions have prompted us to revise our analyses, especially with respect to generalizability and intrinsic dimension. Below, we provide a point-by-point response.
>
> # Major topics (1/2)
>
> > The generalizability claim is quite bold, and ought to be backed up with more data. The authors only show one held-out subject, but then claim that the method generalizes across subjects (plural). I would appreciate a complete leave-one-out analysis, leaving out each subject in turn and quantifying the change in classification accuracy with & without subject N.
> - We agree with the reviewer and we have done this analysis, as suggested. A full hold-one-out analysis can now be seen in the revised text; see revised text in Figure 4. A full break-down by task can be found under “Appendix N: Supplementary Results”.
> - The results hold up, the held-one-out results are consistent with the previous conclusions  and more thoroughly demonstrate generalization. In all cases, the BrainBERT representations result in far better decoding accuracy than the linear baseline.
>
> > This claim is also made when using the top 10 electrodes across all subjects, and the top 10 electrodes for the held-out subject. To support the generalizability claim across electrodes, it would be useful to see where these electrodes are. Does the model still generalize if the electrodes are in quite different locations?
> - The locations of these electrodes can now be seen in figure 9 under “Appendix L: Supplementary figures”. In general, it can be seen that the selected electrodes cover a wide variety of positions.
>
> > The "few-shot learning" claim is also too strong -- the model does require fewer training samples, and may be within the range of other few-shot learning claims in LM papers (which are typically some fraction of the original training set and represent many labeled samples, rather than computer vision papers which are just 2-5 samples per class). However, it definitely does not fall into the range the authors suggest is crucial for many clinical and BCI studies where, in the authors' words, "subjects may participate in only a few dozen trials"
> - BrainBERT exceeds the performance of a linear classifier using only around 50 trials, as shown in figure 5. This is why we discuss BCI where “subjects may participate only in a few dozen trials”. It is true that BrainBERT can take advantage of the availability of more data, but this is a strength: quick short-term performance + high long-term capacity. We also note that numerous computer vision publications refer to 50-shot learning as few-shot learning, this is to an extent subjective as no formal definition exists; we are, at in any case, at the low end of the spectrum in terms of number of examples needed and we reduce the number significantly while gaining performance. But to ensure that there is no confusion, the section title is now "data efficiency."
>
> > The authors suggest that there is a clustering of high ID electrodes in motor areas. This does not appear to be true to me -- no more than, say, left lateral PFC, which is where I see a cluster -- perhaps the authors can circle the area they are referring to in Fig. 5.
> - Thank you for pointing this out; it has prompted us to revise our analyses. Our initial purpose was to identify the locations where high ID activity was occurring. To do this, we averaged over the electrode ID per region, which unfairly penalized larger regions that covered both high and low ID areas.
> - In our revised analysis, we restrict our attention to the electrodes in the top-10th percentile. Those electrodes, which we have now circled in the figure as suggested, are mainly located in the left frontal and temporal lobes, as the reviewer observed. Among these high ID electrodes, the regions with the highest ID are the supramarginal gyrus, the lateral orbitofrontal cortex, and the amygdala. We’ve revised the text in “Section 4: Intrinsic Dimension” and a complete ranking can be found in “Appendix I: Intrinsic Dimension.”

---

> > ### Author Response · Authors · 2022-11-16
> > **Response to reviewer suggestions**
> >
> > # Major topics (2/2)
> > > Second, this is not actually task agnostic. All the data used to train their model is from movie viewing, and they may have seen extremely different ID numbers for other tasks (e.g. motion tasks that are not just passive perception).
> > - We use “task” in the machine learning sense, as the target for a model, i.e., decoding loudness from the neural data. BrainBERT, like BERT, is task-agnostic in the sense that it learns generic representations which are useful for downstream classification of volume, pitch, and linguistic features without any of these features being provided as labels during pretraining. We’ve clarified this point in the text under “Section 4: Intrinsic Dimension”.
> >
> > > What is the interpretation for a high ID area, and what is unexpected about these two regions having a high ID?
> > - First, we would like to clarify that our intention is not to present unexpected neuroscience findings in this work, but to introduce a new methodology and provide a proof of concept for the types of studies that might be conducted in the future. In the analysis discussed above, we have found high ID areas in the frontal and temporal lobes, particularly in areas that have been implicated in phonological processing (supramarginal gyrus [1]) and emotion (amygdala [2]). The high intrinsic dimension of these areas suggests that these regions visited a greater number of distinct states over the course of the movie viewing.
> > - For comparison, studies of the intrinsic dimension of CNN layers find that for an image classification task, the ID increases and then decreases as a function of network depth [3]. That is, high ID corresponds with intermediate preprocessing done by the hidden layers. We’ve added text under “Section 4: Intrinsic dimension” to clarify this.
> >
> > > The authors suggest that "embeddings of the neural data provide a new means by which to investigate the brain", but they fall short of illustrating how a neuroscientist would do this. What is a conclusion one could draw from the type of data presented here?
> > - We’re excited by the many possible applications, including: using the embedding space to build a metric for functional connectivity as simple correlations are used today. Unsupervised approaches to data exploration, like understanding how intrinsic dimensionality changes during sleep, might shed light on the fine-grained mechanisms behind sleep. Simply increasing the raw performance of classifiers with these embeddings could have high impact for clinical tasks such as seizure detection. Moreover, the NLP community has found numerous creative applications for such embeddings that were not imagined by their original authors; we expect the same to happen in the neuroscience community and are following up on this.
> >
> > > A neuroscientist might ask why one should use BrainBERT embeddings instead of training decoders directly on neural data.
> > - We should have made this clear in the manuscript and will update it. Here are a few reasons why neuroscientists would want to use BrainBERT instead of training decoders on neural data:
> >  1. A performance boost. Results in neuroscience are often marginal, performance is low enough that one wonders if the experiment will replicate. Taking a classifier that has 55% accuracy and turning it into one with 65% accuracy removes this doubt.
> >  2. Sample efficiency. Neural recordings are labor and cost intensive so they are almost always in short supply. Being able to get the same results with 50 trials instead of 300 trials means that we can save time, money, and design better experiments (with more controls or conditions).
> > 3. Decoding that which might not have been possible before. This is a combination of 1 and 2. A classifier that might have had chance performance on the raw data may perform well on BrainBERT embeddings. We reduce the rate of false negatives.
> > 4. Faster turnaround time on experiments. This is again a consequence of 1 and 2. With a weak method one needs to accumulate significant data before the results are statistically meaningful. This means investing significant time into an experiment that may well fail. A more powerful method like BrainBERT cuts down on this loop.
> > - This is of course in addition to other tasks that can be performed with the embeddings themselves.
> >
> > > Other authors have claimed that using NN models of the brain provide the ability to perform causal interventions (on the model, instead of the brain) ... Causal interventions can actually be done with iEEG experiments -- so what else might BrainBERT provide?
> > - This is a good point! We have such data related to causal interventions and their behavioral effects. But do not use it in this initial publication as it introduces significant additional complexity. BrainBERT might be usable to predict the effects of interventions, maybe by inverting it with a diffusion model. We had not thought about the connection between the two, but will do so in the future. Thanks!

---

> > > ### Author Response · Authors · 2022-11-16
> > > **References**
> > >
> > > # References
> > > [1] Deschamps, Isabelle, Shari R. Baum, and Vincent L. Gracco. "On the role of the supramarginal gyrus in phonological processing and verbal working memory: evidence from rTMS studies." Neuropsychologia (2014)
> > >
> > > [2] Gallagher, Michela, and Andrea A. Chiba. "The amygdala and emotion." Current opinion in neurobiology (1996)
> > >
> > > [3] Ansuini, Alessio, et al. "Intrinsic dimension of data representations in deep neural networks." Advances in Neural Information Processing Systems (2019)

---

> > > ### Comment · Reviewer_x1NZ · 2022-11-23
> > > **Appreciate the thorough revision / reply**
> > >
> > > I thank the authors for their thorough reply. They have addressed both major and minor concerns, and I will therefore be increasing my score to an 8. In particular, I am more convinced by (1) their leave-one-out subject results, and (2) their more accurate and reasonable discussion of the intrinsic dimensionality analysis.
> > >
> > > I would like to reply to the comments on why a neuroscientist should use BrainBERT embeddings. I think a more in-depth discussion on this is probably out of the scope of this paper, which is mostly about developing the ML method and not about understanding the brain. I do want to point out that if something can be linearly decoded from BrainBERT, but not directly from the neural recordings, this may not clarify any biological mechanism by which something is represented or computed. Instead of studying the brain to answer this question, neuroscientists would need to study BrainBERT itself! This may be useful -- a model can certainly be probed in more ways than a biological organ. However, other neuroscientists may be highly skeptical of this approach. Regardless, I still think this is an interesting contribution that may lead to the community making advances on this question.

---

> ### Author Response · Authors · 2022-11-16
> **Response to reviewers minor questions and suggestions**
>
> # Minor topics and questions
> > It is difficult to compare performance on the decoding tasks across some of the manipulations. The averages are reported in the text, but it's not clear what the authors are averaging over since this is not reported in the table. Please include the averages in the tables as well.
> - Done. The averages, which are computed across tasks, have been added as a column in the tables.
>
> > Authors also report some variance in performance (e.g. AUC +/- value), but unclear what this represents -- e.g. standard deviation over 3 random seeds? Please clarify.
> - In tables 1 and 2, the variance reported is calculated with respect to a fixed set of electrodes. For a given task, this subset is selected by first training a linear classifier over /all/ electrodes. We find the top 10 electrodes with the highest linear classifier performance. This subset is then held fixed over all comparisons between the baselines, ablations, and our model. We’ve added text under “Section 3: Baselines” to clarify this.
> - In figure 5, error bars show 95-percent confidence over 3 random seeds. The caption now reflects this.
>
> > It seems the authors are only using the last layer output of the frozen BrainBERT to train the decoders. Is this correct? If so, I'm curious how different layer outputs would change the performance across tasks.
> - Yes, we are using the last output. The reviewer raises an interesting question. We have now computed the decoding performance of the frozen BrainBERT features per layer of the transformer encoder stack. We see that for all tasks, performance increases with depth, peaking at the second to last layer. This can now be seen under “Appendix M: Supplementary Results”.

---

### Official Review · Reviewer_98Er · 2022-10-25

**Confidence:** 4
**Correctness:** 4
**Technical Novelty And Significance:** 3
**Empirical Novelty And Significance:** 4
**Recommendation:** 8

**Clarity, Quality, Novelty And Reproducibility:**

Clarity: The paper is written very clearly. Almost all the information presented is easy to follow
Quality: High
Novelty: The method is not novel as it is based on BERT. However, the application, results are new.
Reproducability: To reproduce this, I hope the authors are planning to release models and data without which it will be difficult.

**Strength And Weaknesses:**

Strengths:
1. The paper is clearly written and easy to follow.
2. The approach is validated on multiple tasks from determining volume level (low-level) to speech onset (high level) , compared with relevant baselines with relevant ablations thus making the evaluation of the proposed approach quite solid.
3. The results are promising given the brain datasets are generally smaller, this approach shows a new direction of feature learning directly using brain recordings and then applying on decoding tasks.

Weaknesses:
1. The only weakness of this work I can see is that it is build on existing approach BERT and does not propose something entirely novel. However, I do not believe it is a major weakness as it is applied to a different problem and modified to take into account different modality.
2. It is not clear to me how 10 best electrodes were selected. I recommend clarifying it and perhaps adding some results with best 50 or best 20 just to give an idea how much impact selection have on the decoding performance.

**Summary Of The Paper:**

This paper proposes BrainBERT , a self-supervised representation learning approach for intracranial recordings inspired by success in NLP and speech recognition.

First, BrainBERT is trained on intracranial recordings while the subjects were watching videos without driven by any particular decoding task. Then, pretrained model is either finetuned or used as a feature for linear decoding of different low-level and high-level tasks.

The experiments show that features learned by BrainBERT
1. Perform better than linear decoders directly trained on recordings
2. Requires less data to reach good decoding performance
3. Generalizes to new subject and electrodes

**Summary Of The Review:**

This paper presented exciting new results by application BERT type training on intracranial recordings. The approach presented show promise and has potential to be applied on several other type of brain recordings to gain more interesting insights about the brain.

Therefore, I recommend acceptance.

---

> ### Author Response · Authors · 2022-11-16
> **Response to reviewer comments**
>
> Thank you for the helpful review! Below, we give a point-by-point response:
>
> > The only weakness of this work I can see is that it is build on existing approach BERT and does not propose something entirely novel. However, I do not believe it is a major weakness as it is applied to a different problem and modified to take into account different modality.
> - A key novelty here is the application of the ideas in BERT to human neurophysiological recordings, focusing on language processing as a key example. We expect that this approach can have an influential impact in the ability to better understand brain representations, better link representations in biological and artificial neural networks, and also in practical applications to decode neural signals across subjects in a variety of domains.
>
> > It is not clear to me how 10 best electrodes were selected. I recommend clarifying it and perhaps adding some results with best 50 or best 20 just to give an idea how much impact selection have on the decoding performance.
> - We selected the top 10 electrodes with the highest linear classifier performance. This ensures that when making comparisons to the linear baseline, we are giving the baseline model the best possible chance. This subset is then held fixed over all comparisons between the baselines, ablations, and our model.  We’ve added text in “Section 3: Baselines” to clarify this.
> Results for the best 20 electrodes can now be seen under “Appendix N: Supplementary Results.” There is a slight change in the exact numbers, but the relative ordering of the BrainBERT models and the linear decoding baseline remains the same.

---

### Official Review · Reviewer_TZqs · 2022-10-26

**Confidence:** 3
**Correctness:** 3
**Technical Novelty And Significance:** 4
**Empirical Novelty And Significance:** 2
**Recommendation:** 8

**Clarity, Quality, Novelty And Reproducibility:**

Quality: The technical advancement is in the proposed modeling framework and the diverse experiments the authors conducted to show the usefulness of this approach.
Clarity: Could be improved significantly, especially for model details like the pertaining loss, performance etc..
Novelty: tied into quality comment above.

**Strength And Weaknesses:**

Strengths: This paper presents a new and interesting computational framework for identification tasks using neural recordings. I think this will be an important research direction going forward and commend the authors for pursuing a novel line of work.

Weaknesses: While the paper was interesting, several of the implementation details were confusing or missing altogether. I highlight a few examples below.

- How does the model handle different electrodes? Is training shared across them? I was also confused by “First, the Linear (5s time domain input) network is trained once per electrode, for all held-out electrodes.” on page 6 under baselines.
    - I am particularly interested in the trade-offs of using fewer electrodes and if the authors found better performance on downstream task based on the location of the electrodes, as would be expected.
- Is there any possibility for the model to encode information in the population code?
- What is the context window during training? I presume 5 seconds from details in the introduction of section 3.
- What do the training curves look like? What is the pretraining performance? How hard was hyper-parameter tuning?
- Can the authors comment on the fact that the decoding accuracy is above the deep NN baseline only when the model is finetuned?
- Nit: I would decrease the curvature contrast in Fig. 5B so the electrode values are more visible.
- I am not sure what to make of the intrinsic dimensionality analysis as the links to working memory and a sensory lobe seem arbitrary. Furthermore, the distribution across cortex doesn’t seem to follow any prior findings on gradients or functional distinctions during speech processing.
- How many hours of data per subject on average?
- What does the prediction head look like?
- Is there any relationship between SNR and subject-transfer abilities (there must be!)

**Summary Of The Paper:**

This work proposes a transformer-based architecture with a masked modeling task to operate on intra-cranial recordings directly. Some of the important components of this model include a content-aware loss to capture the burst-like nature of neural activity, a superlet representation of the raw neural data and a masking procedure based on removing frequency/time bands. The authors show that using this pretrained architecture leads to better downstream identification performance on a suite of tasks like detecting speech onsets, distinguishing between speech and non-speech sounds etc..that cannot be attributed to the complex architecture of the model alone. They also look at the impact of using frozen vs. trainable encoding layers on downstream accuracy, generalizing to new subjects and accuracy as a function of training examples. Lastly, they apply PCA to the hidden states of each electrode input to compute the electrode’s intrinsic dimensionality.

**Summary Of The Review:**

Overall, I think this paper is a promising methodological advancement over linearized encoding models. However, I thought the paper was a bit confusing to read and found some of the empirical results to be weak (effective dimensionality).

---

> ### Author Response · Authors · 2022-11-16
> **Response to questions and suggestions**
>
> We thank the reviewer for their helpful suggestions. We’ve revised the manuscript to reflect the suggested clarifications. Below, we give a point-by-point response to the reviewer’s questions.
>
> # Response to questions (1/2)
> > How does the model handle different electrodes? Is training shared across them?
> - Yes. In pretraining, recordings from all electrodes and from all subjects are collected into a single dataset. We now clarify this in “Section 2: Data”.
>
> > I was also confused by “First, the Linear (5s time domain input) network is trained once per electrode, for all held-out electrodes.” on page 6 under baselines
> - By “held out electrodes” we mean the electrode recordings from the held out sessions. We’ve revised the text in “Section 3: Baseline” to clarify this.
>
> > I am particularly interested in the trade-offs of using fewer electrodes and if the authors found better performance on downstream tasks based on the location of the electrodes, as would be expected.
> - Our goal is to make one large network that generalizes to unseen electrode locations in unseen subjects. This means that we can pool together as many electrodes as we want from many different recording sessions. Thus, computing performance as a function of the pretraining set size has not been our focus. Instead, we try to minimize the amount of training data per-subject per-electrode, in other words, we want future users to use as little data as possible while relying maximally on the efforts of previous users; just like BERT and other language models in NLP.
> - Better performance on the downstream task is dependent on which electrode we are trying to perform the task on, but not on the training set locations.
>
> > Is there any possibility for the model to encode information in the population code?
> - Yes and no. The answer hinges on what the reviewer means by the term “population code”, which has been used to mean different things. Inputs to the model are independent per electrode. Electrodes of course capture the average activity of many neurons. In so far as this is a population code in the frequency domain, the answer is yes. For a population code one might also want to use multiple electrodes per input, which does not happen in the model as is. We could in principle concatenate neighboring electrodes and create such population codes. It would be exciting to do so in a future extension of the model.
>
> > What is the context window during training? I presume 5 seconds from details in the introduction of section 3.
> - Correct; it is 5s. We've added text in "Section 2: Data" to make this more clear.
>
> > What do the training curves look like? What is the pretraining performance? How hard was hyper-parameter tuning?
> - We’ve added the pretraining curves and exemplar reconstructions under “Appendix L: Pretraining performance”.
> - We manually chose a few parameters (learning rate, context aware weighting, mask width) to try since the model takes half a week or more to train on our hardware. It’s likely that systematically tuning the hyperparameters rather than generally guessing would result in much higher performance.
>
> > Can the authors comment on the fact that the decoding accuracy is above the deep NN baseline only when the model is finetuned?
> - Note that this is only the case for (2/4) of the tasks. For the speech vs. non-speech and volume classification, frozen embeddings give better accuracy.
> - The deep NN baseline is very powerful! More powerful than the linear classifiers that the vast majority of publications in neuroscience use. In a sense, the deep NN baseline should always beat BrainBERT if there is enough data. Likely a large model with more pretraining (we are working on scaling up) would bridge this gap. The fact that BrainBERT is competitive with the deep NN approach shows its significant value.
>
> > Nit: I would decrease the curvature contrast in Fig. 5B so the electrode values are more visible.
> - Thanks! The figure has been updated.

---

> > ### Author Response · Authors · 2022-11-16
> > **Response to questions (cont.)**
> >
> > # Responses to questions (2/2)
> > > I am not sure what to make of the intrinsic dimensionality analysis as the links to working memory and a sensory lobe seem arbitrary. Furthermore, the distribution across cortex doesn’t seem to follow any prior findings on gradients or functional distinctions during speech processing.
> > - Thank you for your comment. The purpose of our initial analysis was intended to identify the locations where high ID activity was occurring. To do this, we averaged over the electrode ID per region, which unfairly penalized larger regions that covered both high and low ID areas. We have revised our analysis and we now restrict our attention to the electrodes in the top-10th percentile.
> > - Those electrodes, which we have now circled, are mainly located in the frontal and temporal lobes. Among these high ID electrodes, the regions with the highest ID are the supramarginal gyrus (phonological processing [1]), the lateral orbitofrontal cortex (sensory integration [2]), and the amygdala (emotion [3]). A complete ranking of all regions can be found in “Appendix I: Intrinsic Dimension.”
> > - We give an interpretation for these results in the revised text: similar studies of the intrinsic dimension of CNN layers find that for an image classification task, the ID increases and then decreases as a function of network depth [4]. That is, high ID corresponds with intermediate preprocessing done by the hidden layers.
> >
> > > How many hours of data per subject on average?
> > - There's approximately 4.3 hours of data per subject. We've added text in "Section 2: Data" to clarify this. A complete account of the amount of data collected per subject can be found in "Appendix J: Dataset Statistics".
> >
> > > What does the prediction head look like?
> > - The spectrogram prediction head consists of two stacked fully connected layers. This can be found in Section 2 under "Architecture". We've also added text under "Appendix G: Pretraining Parameters'' for completeness.
> >
> > > Is there any relationship between SNR and subject-transfer abilities (there must be!)
> > - We agree with the reviewer, generally speaking, that there must be a relationship between SNR and subject-transfer ability. But it’s important to note that in the context of our intracranial recordings, there is not a guaranteed, completely reliable way to distinguish between what is signal and what is noise.
> > - However, we implemented several preprocessing steps to remove artifacts and thus improve SNR, including filtering out line noise, Laplacian re-referencing, and rejecting obviously noisy electrodes (see Appendix F: Preprocessing). Moreover, all data was collected in the same hospital using the same instruments, which further minimizes the inter-subject variance in SNR. This question likely can’t be answered with our data.
> >
> > ## References
> > [1] Deschamps, Isabelle, Shari R. Baum, and Vincent L. Gracco. "On the role of the supramarginal gyrus in phonological processing and verbal working memory: evidence from rTMS studies." Neuropsychologia (2014)
> >
> > [2] Rolls, Edmund T. "Convergence of sensory systems in the orbitofrontal cortex in primates and brain design for emotion." The Anatomical Record Part A: Discoveries in Molecular, Cellular, and Evolutionary Biology: An Official Publication of the American Association of Anatomists (2004)
> >
> > [3] Gallagher, Michela, and Andrea A. Chiba. "The amygdala and emotion." Current opinion in neurobiology (1996)
> >
> > [4] Ansuini, Alessio, et al. "Intrinsic dimension of data representations in deep neural networks." Advances in Neural Information Processing Systems (2019)

---

> ### Author Response · Authors · 2022-12-08
> **Follow up**
>
> The end of the reviewing period is approaching, and we wanted to check if you had a chance to read our response below. If you have any further questions, please don't hesitate to let us know!

---

### Decision · Program_Chairs · 2023-01-20

**Decision:**

Accept: poster

**Justification For Why Not Higher Score:**

As raised by the reviewers, this dataset is small, and its not clear how it would generalize to new subjects or subjects doing a different task.  A more through evaluation would help this paper significantly.

**Justification For Why Not Lower Score:**

The reviewers were convinced by the rebuttals, and the two highest scores are also the most confident.  I know the work of one of the high-scoring reviewers and so trust their evaluation.

**Metareview: Summary, Strengths And Weaknesses:**

This paper details a method for applying transformers to brain imaging data, specifically ecog recordings.  The authors show that their method creates representations of brain imaging data that improve downstream tasks.  The reviewers did have some questions about the details and validity evaluation, but were convinced by the rebuttals.  There remain some questions about how this work could inform neuroscience research, and I hope the authors will incorporate those additional thoughts into their manuscript.



Side note from AC: in figure 5 it's quite hard to tell the difference between the purple and blue lines, thicker lines would help.  Also, moving the brain and legend outside of the figure would allow you to increase the size (including the font in the legend as it's currently quite small).  The way the paper is currently laid out there's lots of horizontal space beside F2 to do this.  The font size on the left hand size of figure 6 is also very small.

**Note From Pc:**

if the above contains the word "oral" or "spotlight" please see: "oral" presentation means -> notable-top-5% and "spotlight" means -> notable-top-25%. As stated in our emails, we are disassociating presentation type from AC recommendations

**Summary Of Ac-Reviewer Meeting:**

This was a borderline-ish paper before I prodded the reviewers to consider the rebuttal.  I felt that the reviewers had properly considered the rebuttal and updated their reviews so we didn't need to meet.